# Elevated Piezo levels cause structural and functional alterations in *Drosophila* garland nephrocytes

Paris Hazelton-Cavill[1,2], Karl K Alornyo[1,2], Michelle Bouchard[1,2] ©, Kristina Schulz[1,2], Tobias B Huber[1,2], Barry Denholm[3] ©, Sybille Koehler[1,2,3] ©

**Podocytes, epithelial cells of the glomerular filtration barrier, are constantly exposed to biomechanical forces. These include hydrostatic pressure and shear stress, which increase during diseases such as hypertension or diabetes. To sense and respond to such changes in their physical environment, podocytes express mechanosensors and mechanotransducers. To deepen our knowledge about renal mechanotransduction mechanisms, we used *Drosophila* nephrocytes. Nephrocytes and mammalian podocytes are highly similar in morphology and molecular make-up of the filtration barrier; thus, nephrocytes are considered the homologue cells to podocytes. In addition, nephrocytes also experience biomechanical forces because of haemolymph movement. Here, we investigated the role of the mechanotransducer Piezo in larval garland nephrocytes. Depletion of Piezo produces only a mild functional phenotype, whereas elevated Piezo levels result in a severe phenotype with functional and morphological disturbances. Increased Piezo levels also cause the accumulation of actin stress fibres, increased Cubilin expression, more acidic vesicles, increased mitochondrial mass and/or activity, and elevated superoxide levels.**

## Introduction

Podocytes are highly specialized cells of the kidney that, together with the fenestrated endothelium and the glomerular basement membrane, form the renal filtration barrier, a size- and charge-selective filter (Pavenstädt et al, 2003). Podocytes form primary and secondary foot processes, which enwrap the glomerular capillaries in the mammalian kidney. Upon injury, podocytes undergo morphological changes, lose contact with the basement membrane, and get lost into the primary urine. This leaves the capillaries blank, as podocytes are post-mitotic and cannot be replaced, resulting in proteinuria and ultimately kidney failure. One cause of podocyte injury, among others, is an increase of biomechanical force experienced by these cells (Endlich et al, 2001; Feng et al,

2018; Koehler et al, 2020; Greiten et al, 2021). Because of their position in the glomerular filtration barrier, podocytes are constantly exposed to biomechanical forces: hydrostatic pressure because of blood pressure in glomerular capillaries, and shear stress because of the filtration of blood into primary urine (Endlich et al, 2017). Both forces increase during diseases such as diabetes and hypertension, ultimately resulting in podocyte loss (Feng et al, 2018). In order to prevent podocyte loss that is induced by increased biomechanical force, mechanotransduction processes need to be characterized in greater detail. This entails processes by which external stimuli such as stretch or force are sensed and transduced into biochemical or electrical signals in the cell, enabling cells to adapt to changes in their physical environment. Such mechanisms could be targeted to develop novel interventions during glomerular disease and podocyte injury.

Investigations into podocyte biology and dysregulated mechanisms during injury are subject to several limitations, such as accessibility of the cells and the lack of suitable tools to study mechanotransduction in greater detail. Thus, the use of alternative models such as cell culture and *Drosophila melanogaster* can be beneficial for the functional assessment of proteins and pathways. Notably, *Drosophila* nephrocytes have been established as the homologue cells of mammalian podocytes, as they not only share similarities in filtration function, but are also highly comparable in morphology and molecular make-up (Weavers et al, 2009; Zhuang et al, 2009). Nephrocytes are part of the excretory system of *Drosophila* together with the Malpighian tubules. They are specialized cells that function as filtration units, capable of filtering toxins and waste products from the haemolymph—the blood equivalent. At their surface, nephrocytes form membrane invaginations resulting in foot processes, which are spanned by the nephrocyte diaphragm and surrounded by a basement membrane (Weavers et al, 2009). They occur in two distinct populations: garland and pericardial nephrocytes, which are similar, but differ slightly in morphology and physiology (Xu et al, 2022; Meyer et al, 2024). Garland nephrocytes are localized around the oesophagus, whereas pericardial nephrocytes lie along the heart tube. Because the heart tube performs peristaltic, rhythmic movements to pump

[1]III. Department of Medicine, University Medical Center Hamburg-Eppendorf, Hamburg, Germany [2]Hamburg Center for Kidney Health (HCKH), University Medical Center Hamburg-Eppendorf, Hamburg, Germany [3]Biomedical Sciences, University of Edinburgh, Edinburgh, Scotland

Correspondence: Sy.koehler@uke.de

 

and circulate haemolymph, the pericardial cells are exposed to biomechanical forces under physiological conditions. Alongside other mechanosensors and mechanotransducers described in *Drosophila*, we and others have confirmed the presence and role of the mechanotransducer Piezo in pericardial cells in adult flies (Schulz et al, 2024; Zhao et al, 2024). Piezo is a mechanosensitive ion channel, which responds to mechanical force and allows the influx of cations such as $Ca^{++}$ and $Na^+$ (Coste et al, 2010, 2012). In mammals, two Piezo proteins are present: Piezo1, which is important for sensing blood flow–associated shear stress and for blood vessel development (Ranade et al, 2014; Cahalan et al, 2015; Retailleau et al, 2015); and Piezo2, which functions in touch and proprioception (Zhong et al, 2018). Recent studies investigated Piezo levels in patient-derived kidney tissue and murine disease models and revealed a dysregulation of Piezo1 and Piezo2 levels (Fu et al, 2024; Ogino et al, 2024; Schulz et al, 2024). In detail, Piezo1 is down-regulated in podocytes isolated from diabetic kidney disease patients (Schulz et al, 2024), but up-regulated in a murine hypertension model and in some forms of lupus nephritis (Fu et al, 2024; Ogino et al, 2024). Piezo2 levels are also dysregulated in glomerular tissue obtained from lupus nephritis patients (Fu et al, 2024). Although some studies have investigated the functional role of Piezo in podocyte and nephrocyte biology, only a limited number of downstream pathways have been assessed. Thus, in this study we further investigated the downstream effects of depletion or overexpression of Piezo on nephrocyte biology.

The effect of a mechanical stimulus on larval garland and adult pericardial nephrocytes was first assessed, confirming that both cell types respond. This response in garland nephrocytes seems to be at least partially mediated by Piezo. We then examined the functional role of Piezo in larval garland nephrocytes and observed morphological and functional phenotypes upon the overexpression of the channel, but only a mild functional phenotype in our Piezo knockout model. To unravel the underlying mechanisms of this functional phenotype, we assessed Cubilin and Cubilin 2 expression—two receptors involved in protein uptake—and could show an elevation of Cubilin levels upon the overexpression of Piezo. Elevated Piezo protein levels also resulted in accumulation of actin fibres. In addition, we find evidence suggesting that the activation of Rho1 might be downstream of Piezo, as simultaneous overexpression of Piezo and a dominant-negative variant of Rho1 reversed the observed filtration phenotype upon Piezo overexpression. The overexpression of the channel caused an increase in mitochondrial mass and/or activity and elevated superoxide levels. Last, we assessed the potential beneficial effect of the unspecific Piezo inhibitor tarantula toxin to prevent Piezo-mediated phenotypes in nephrocytes.

# Results

### *Drosophila* nephrocytes respond to a biomechanical stimulus with decreased FITC-Albumin uptake

Because of their position along the heart tube, pericardial nephrocytes are exposed to biomechanical force under physiological conditions, as the heart continuously pumps haemolymph out of the ostia next to pericardial cells (Fig 1A). The second nephrocyte population, the garland cells, lie around the oesophagus and are floating in the haemolymph (Fig 1A). It is conceivable that larval garland nephrocytes also experience biomechanical force because of haemolymph circulation generated by the symmetrical peristaltic contractions that enable larval movement. In contrast to pericardial cells, garland nephrocytes are not attached to tissue but are floating in the haemolymph, thus potentially experiencing different biomechanical forces. To assess whether both nephrocyte populations respond to biomechanical force, we applied a mechanical stimulus ex vivo and investigated larval garland and adult pericardial nephrocyte function. In detail, the SYS-PV830 microinjector was used to apply a pressure-controlled release of buffer into a glass dish containing the tissue (Fig 1B). This approach mimics shear stress and has been used in our previous study (Schulz et al, 2024). Of note, this ex vivo experiment might not resemble physiological conditions experienced by floating nephrocytes in vivo. In our setup, buffer is released from the pressure injector at 0.5 bar; however, the pressure experienced by nephrocytes is reduced because of dissipation over distance. The injection needle is positioned ~1 cm away from the glass dish and the surface of the buffer in which the dissected tissue is floating. Importantly, we have previously used this approach to study $Ca^{++}$ influx in the absence and presence of Yoda1, as well as the effects of Yoda1 on nephrocyte biology [11]. In these experiments, mechanical stimulus was also applied to control (DMSO-treated) cells and no morphological or functional phenotypes were observed in larval garland nephrocytes. Together, these observations suggest that even if the applied pressure is higher than physiological levels, we do not induce severe damage through mechanical stress.

Here, applying a stimulus of 0.5 bar for 3 s on adult pericardial nephrocytes revealed a significant reduction of FITC-Albumin uptake, confirming a response of these cells to a mechanical stimulus (Fig 1D). We repeated this experiment in isolated larval garland nephrocytes, which also showed a significant decrease in FITC-Albumin uptake after applying the stimulus (Fig 1C). The FITC-Albumin uptake assay is a widely used assay to assess nephrocyte function (Hermle et al, 2017). FITC-Albumin can pass through the nephrocyte diaphragm and bind to receptors such as Cubilin and Amnionless, which are located in the lacuna membrane. After binding to these proteins, FITC-Albumin is endocytosed before degradation in vesicles including the lysosomes. In addition to detecting changes in nephrocyte uptake function, the FITC-Albumin assay can also indicate the occurrence of morphological alterations such as loss of nephrocyte diaphragm and reduced lacuna channels, which could result in a reduced number of receptors available for uptake (Hermle et al, 2017). Thus, incubation of nephrocytes with FITC-Albumin can help identify alterations in endocytosis and nephrocyte diaphragm integrity, as well as lacuna structure. Of note, both decreased and increased FITC-Albumin levels can point towards morphological alterations or changes in lysosomal degradation, as loss of nephrocyte diaphragms can also allow for easier access to receptors such as Cubilin (Dow et al, 2022).

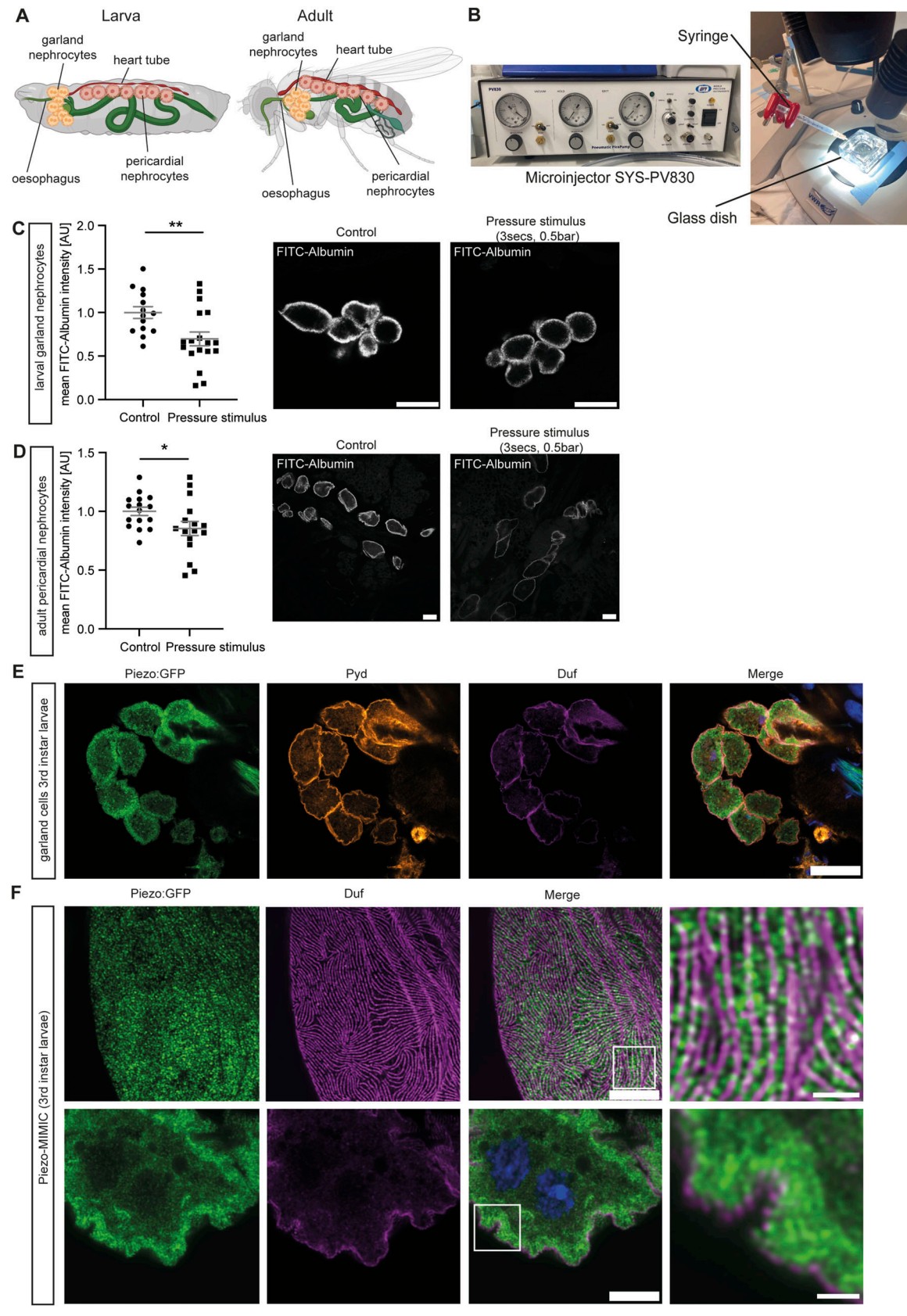

Our data suggest that although larval garland nephrocytes float in haemolymph and thus might experience different biomechanical forces in vivo when compared to pericardial cells, they respond to biomechanical stimuli similar to pericardial nephrocytes. Prior to functional assessment of mechanotransduction in larval garland nephrocytes, we analysed the expression of one potential candidate involved in these processes, the mechanotransducer Piezo. Searching the fly kidney atlas revealed the expression of Piezo in both cell types (garland nephrocytes: 36.6% of cells; pericardial nephrocytes: 18.3% of cells) (Xu et al, 2022). Of note, adult flies were used within this study. For further support of Piezo expression in larval garland nephrocytes, we used a recently published Piezo MI{MIC} fly strain (Zhao et al, 2024). This fly strain contains a GFP-tag in the endogenous Piezo sequence and hence labels endogenous Piezo. Analysis of Piezo-GFP MIMIC flies revealed that most of the cells positive for Duf (nephrocyte marker) also express Piezo under physiological conditions. Co-localization experiments using the nephrocyte diaphragm proteins Duf and Pyd revealed that Piezo is expressed at the cell cortex close to the nephrocyte diaphragm and on the cell surface of larval garland nephrocytes (Fig 1E and F). Zhao et al reported a fingerprint-like pattern on the surface using the same Piezo MI{MIC} fly strain in adult pericardial cells, which is similar to our observation in larval garland cells. This further confirms the feasibility of this approach and the expression of Piezo in larval garland cells (Zhao et al, 2024). Interestingly, Zhao et al also investigated the localization of overexpressed Piezo-GFP, which showed expression at the cell surface, but also in the lacunae, which are membrane invaginations (Zhao et al, 2024). This is in line with our observations in Piezo-overexpressing nephrocytes in which Piezo also seems to localize into the lacuna system (Figs S2C and S3A, C, and E).

Taken together, our data strengthen the hypothesis that larval garland nephrocytes sense biomechanical force and respond to such alterations in their physical environment. Whether the mechanotransducer Piezo is involved in this process is unknown. Therefore, within this study we investigated its functional role in larval garland nephrocytes in greater detail.

### Loss of Piezo in garland nephrocytes causes a mild functional phenotype

Previous results from our group and others have revealed a severe nephrocyte phenotype associated with both Piezo depletion and overexpression in adult pericardial nephrocytes (Schulz et al, 2024; Zhao et al, 2024). However, data from the fly kidney atlas and a recently published transcriptome of adult garland and pericardial nephrocytes reveal equal expression levels of Piezo in both cell types (Xu et al, 2022; Meyer et al, 2024). Also, our own data show that

both larval garland and adult pericardial nephrocytes respond to a mechanical stimulus ex vivo. Based on these findings, we tested whether the loss of Piezo has an effect on larval garland nephrocytes. We did not observe any morphological changes (Fig 2A) in larval garland nephrocytes using a whole-body knockout (Piezo[KO]) of Piezo, whereas FITC-Albumin uptake was significantly increased (Fig 2C). To account for side effects mediated by the whole-body knockout, we repeated morphological and functional assessment in flies with RNAi-mediated nephrocyte-specific Piezo knockdown. This also produced no morphological phenotype and, in contrast to the whole-body knockout, did not result in a FITC-Albumin uptake phenotype (Fig 2B and D). To confirm knockdown efficiency, we performed FISH and observed a significant decrease of Piezo-specific mRNA in garland nephrocytes (Fig S1A). Moreover, to highlight the importance of Piezo in nephrocyte biology, we analysed the effect of applying a mechanical stimulus in the absence of Piezo. This revealed no significant FITC-Albumin uptake defect upon loss of Piezo and a partial rescue of the phenotype observed in control cells exposed to a mechanical stimulus. These data suggest that the FITC-Albumin uptake phenotype exhibited after a mechanical stimulus is at least partially mediated by Piezo. Given that nephrocytes depleted of Piezo show a slight decrease in FITC-Albumin uptake after the application of a mechanical stimulus, we conclude that other mechanotransducers likely also play a role in the development of this phenotype (Fig S1B).

Taken together, loss of Piezo induced by a whole-body knockout resulted in a mild phenotype in larval garland nephrocytes with an increased FITC-Albumin uptake. Whether this phenotype results from effects mediated by the whole-body depletion remains unclear for now. It is also feasible that the RNAi-mediated knockdown is not as strong as the knockout to induce the same phenotype observed in the whole-body knockout.

### Elevation of Piezo wild-type levels causes a nephrocyte phenotype

Next, we assessed the effects of elevated Piezo levels in larval garland nephrocytes, as it has been previously shown that Piezo levels can be either down- or up-regulated in different glomerular diseases contributing to pathological effects (Fu et al, 2024; Ogino et al, 2024; Schulz et al, 2024). The UAS-Gal4 system used here is temperature-sensitive, resulting in increased expression levels at higher temperatures (Duffy, 2002). By comparing the expression levels of Piezo wild type through assessment of FLAG intensity at 25°C and 28°C, we confirmed the temperature-dependent increase in Gal4 efficiency (Fig S2C). To assess the effect of two different expression levels of Piezo, we kept flies at 25°C (medium expression) or 28°C (high expression). Interestingly, only the higher

**Figure 1.** ***Drosophila* nephrocytes respond to a biomechanical stimulus with decreased FITC-Albumin uptake.**
**(A)** Schematic of larvae and adult flies highlighting the localization of garland nephrocytes around the oesophagus and the pericardial cells along the heart tube. Images created with BioRender.com. **(B)** Setup to apply a controlled mechanical stimulus. The SYS-PV830 microinjector is connected with a 1 ml syringe with a 26G needle and filled with haemolymph-like buffer (HL3.1). Tissue is placed within the glass dish under the microscope. Buffer is released in a controlled way: 0.5 bar for 3 s. **(C)** Larval garland cells exposed to a mechanical stimulus respond with a significant decrease of FITC-Albumin uptake. Experiments were done with Oregon R (OreR) wild-type third instar larvae. *t* test: **$P < 0.01$. Scale bar = 25 $\mu m$. **(D)** Adult pericardial nephrocytes show a significant decrease of FITC-Albumin uptake after applying a mechanical stimulus. Flies used are from the OreR strain and 1- to 3-d-old. *t* test: *$P < 0.05$. Scale bar = 25 $\mu m$. **(E, F)** Immunofluorescence staining highlights the nephrocyte diaphragm in larval garland cells by visualizing Duf (dNEPH) and Pyd (dZO1) together with Piezo (Piezo:GFP; Mi{MIC} line 60209). GFP was enhanced using an anti-GFP antibody. This reveals expression of Piezo at the cortical region and at the cell surface. **(E, F)** Scale bar = 25 $\mu m$ (E) and = 5 and 1 $\mu m$ (magnification) (F).

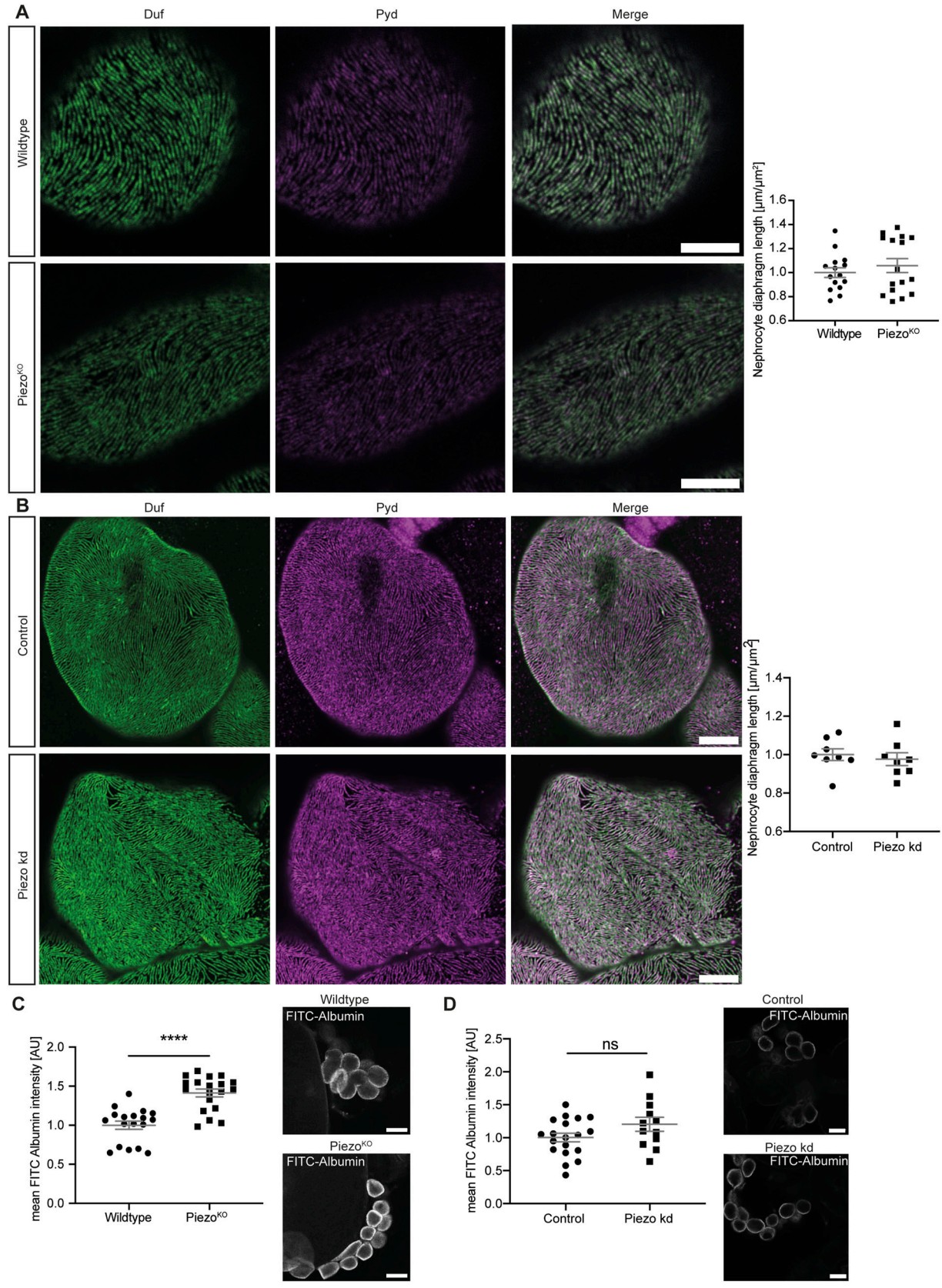

expression levels of Piezo wild type (28°C) caused a nephrocyte phenotype with altered nephrocyte diaphragm density and increased FITC-Albumin uptake (Figs 3A–D and S2A and B). Assessment of a Piezo mutant lacking the mechanosensitive domain (Piezo.2306MYC.FLAG) revealed the absence of any phenotype at both 25°C and 28°C, suggesting that the observed phenotype upon overexpression of Piezo wild type is mediated by the mechanosensitive domain of Piezo (Figs 3A–D and S2A and B). We confirmed the expression of both Piezo variants in larval garland nephrocytes by visualizing the nephrocyte diaphragm and Piezo via antibody-mediated staining against the FLAG-tag (Fig S3A–F). FLAG-tagged Piezo localized to the cell cortex and into the lacuna system, similar to what has previously been described for adult pericardial nephrocytes (Schulz et al, 2024; Zhao et al, 2024).

Taken together, the overexpression of Piezo results in a severe nephrocyte phenotype, including morphological and functional disturbances, which seem to be mediated via the mechanosensitive domain of Piezo. Given that the phenotype observed here is more severe when compared to our depletion models, elevated Piezo levels might exert a more detrimental effect on larval garland nephrocytes.

### Overexpression of Piezo results in increased Cubilin levels, more acidic vesicles, and less haemolymph clearance

Our data reveal an increase in FITC-Albumin uptake upon both loss and overexpression of Piezo. To unravel how this might be mediated and whether the involved mechanisms are different, we investigated various proteins and structures in greater detail. Notably, analogous to our observations, similar phenotypes upon depletion and overexpression of Piezo have been observed in *Drosophila* wing discs, presumably resulting from different underlying mechanisms (Mim et al, 2024). The increased uptake of FITC-Albumin in nephrocytes has been attributed either to alterations in lysosomal processing or in access to Cubilin and Cubilin 2 resulting from morphological changes (Dow et al, 2022). Both Cubilin and Cubilin 2 have previously been reported to be involved in receptor-mediated albumin uptake and endocytosis in nephrocytes (Zhang et al, 2013; Atienza-Manuel et al, 2021). Here, we assessed only the whole-body knockout and overexpression flies at 28°C, as these flies present with a significant FITC-Albumin uptake phenotype. This phenotype might be at least partially the result of altered vesicle formation and processing, or a change in receptor expression and localization. Indeed, visualization of Cubilin and Cubilin 2 revealed a significant increase of Cubilin levels in garland nephrocytes overexpressing Piezo wild type (Fig 4A), whereas Cubilin 2 levels remain unchanged (Fig 4B). In contrast, assessment of Piezo[KO] garland nephrocytes did not reveal any alterations in Cubilin or Cubilin 2 expression (Fig 4C and D).

Cubilin and Cubilin 2 can be retained in the endoplasmic reticulum under certain conditions (Udagawa et al, 2018). In our stainings, we observed intracellular localization as well; hence, we tested whether Cubilin and Cubilin 2 translocate to the cortex in our genetic models. Quantification of cortical Cubilin and Cubilin 2 levels revealed a significant increase of Cubilin only in Piezo wild type–overexpressing nephrocytes, which also have an increase of overall Cubilin levels, suggesting no translocation phenotype, especially of Cubilin 2 (Fig S4A and B). Cubilin 2 levels are unchanged upon overexpression of Piezo wild type, which is unexpected as Cubilin and Cubilin 2 function in a complex together with Amnionless (Zhang et al, 2013; Atienza-Manuel et al, 2021). The potential involvement of downstream signalling such as glycosylation and, thus, ER retention cannot be excluded and should be assessed in follow-up studies. However, the increased Cubilin levels might be at least partially responsible for the elevated uptake of FITC-Albumin. Our assays do not reveal whether Cubilin accessibility is indeed facilitated upon morphological alterations in Piezo wild-type overexpression cells.

Given that alterations in lysosomal processing are described as being potentially causative for an increased FITC-Albumin uptake, we visualized acidic vesicles in nephrocytes using LysoTracker. This revealed a significant increase of acidic vesicles in both Piezo-depleted nephrocytes and cells overexpressing Piezo wild type (Fig 4E and F). This increase might be a result of extensive uptake of particles from the haemolymph—similar to the observed increase of FITC-Albumin uptake, which is subsequently degraded within nephrocytes. As LysoTracker dyes cannot distinguish between lysosomes, late endosomes, and other acidic vesicles, we also assessed the presence of Rab7, a late endosome marker. This showed no significant differences upon overexpression or depletion of Piezo, suggesting that the differences observed with LysoTracker are not primarily mediated by late endosomes (Fig S4D).

To understand the involvement of Piezo in haemolymph clearance in greater detail, we next performed silver nitrate (AgNO$_3$) toxin assays. During larval development, nephrocytes are crucial for clearing the haemolymph of toxins and waste products to ensure normal development. Feeding larvae AgNO$_3$ leads to its accumulation in the haemolymph where it must be cleared by nephrocytes to prevent a developmental delay (pupation delay). An impaired nephrocyte filtration/uptake function will result in reduced clearance efficiency and consequently a developmental delay of larvae, which can be used as an indirect way to assess filtration/uptake function. Interestingly, the overexpression of Piezo wild type already resulted in a developmental delay under basal conditions, which became stronger upon feeding AgNO$_3$ (Fig 4G). To exclude an impact of protein overexpression in nephrocytes on haemolymph clearance in general, we repeated the AgNO$_3$ assay with larvae overexpressing GFP in nephrocytes, which did not result in a phenotype (Fig S4C). This effect on

---

**Figure 2. Loss of Piezo in garland nephrocytes causes a mild functional phenotype.**
**(A)** Immunofluorescence staining with Duf and Pyd antibodies does not reveal any morphological changes in Piezo[KO] (w;Piezo[ko];+) larval garland nephrocytes in comparison with control cells (wild type w[1118]). Scale bar = 5 μm. *t* test: ns. **(B)** Visualizing the nephrocyte diaphragm does not show any morphological alterations upon loss of Piezo induced by RNAi. Scale bar = 5 μm. *t* test: ns. Control: w;sns-Gal4/+;UAS-dicer2/+; Piezo kd: w;sns-Gal4/UAS-piezo-RNAi;UAS-dicer2/+. **(C)** FITC-Albumin uptake assays reveal a functional defect with an increased FITC-Albumin uptake in Piezo[KO] larval garland nephrocytes. Scale bar = 25 μm. *t* test: ****P < 0.0001. **(D)** Loss of Piezo induced by RNAi does not cause significant changes in FITC-Albumin uptake. Scale bar = 25 μm. *t* test: ns.

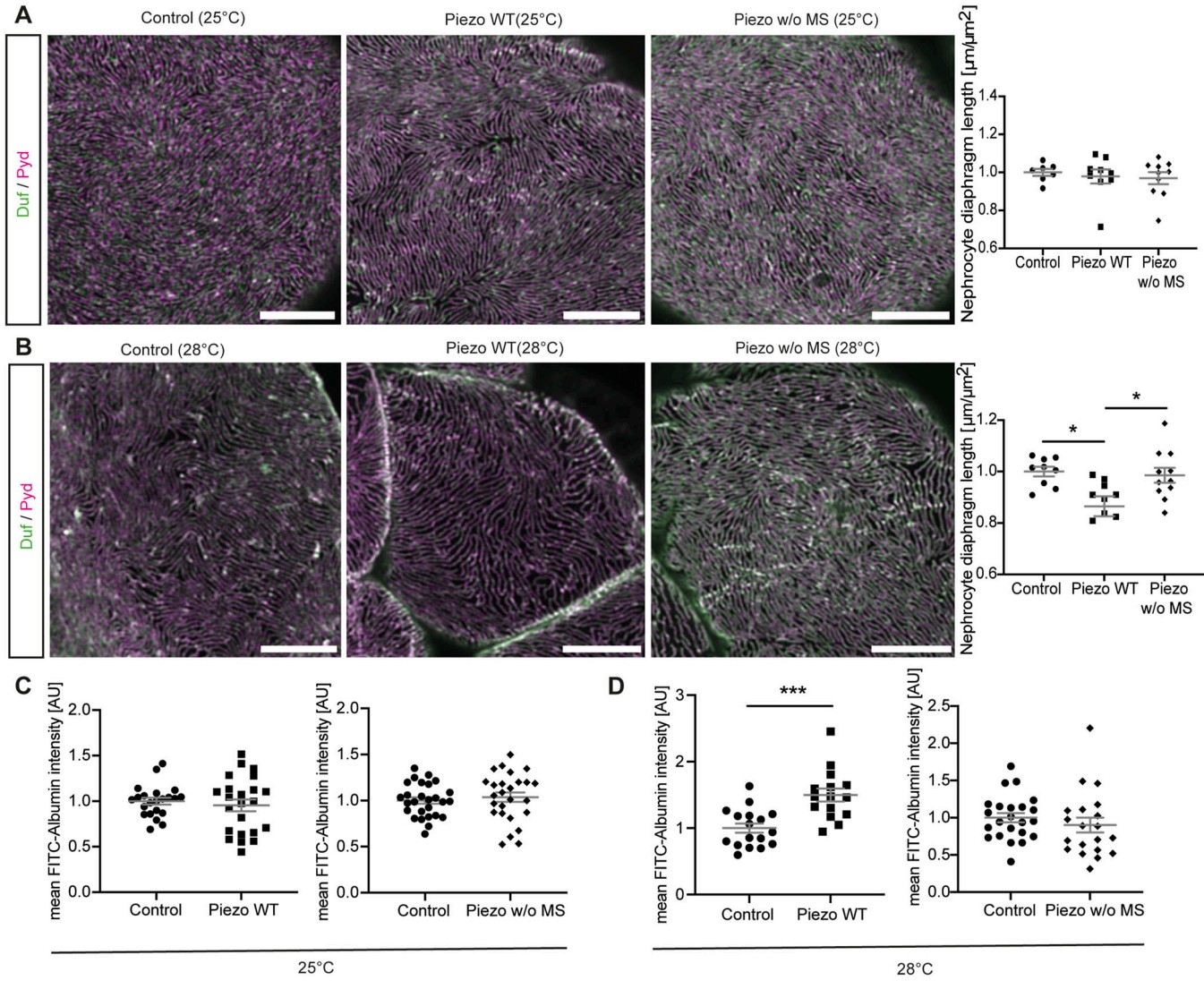

**Figure 3. Elevation of Piezo wild-type levels causes a nephrocyte phenotype.**
**(A)** Immunofluorescence staining with Duf and Pyd antibodies does not reveal any morphological changes in larval garland nephrocytes expressing Piezo wild type (Piezo WT) or a mutant absent of the mechanosensitive channel domain (Piezo w/o MS) at 25°C (medium levels). Green = Duf; magenta = Pyd. Scale bar = 5 µm. Control: w; sns-Gal4/+;UAS-dicer2/+. Piezo WT: w; sns-Gal4/+;UAS-dicer2/UAS-Piezo-FLAG. Piezo w/o MS: w; sns-Gal4/+;UAS-dicer2/UAS-Piezo-2306MYC.FLAG. **(B)** Visualization of the nephrocyte diaphragm with Duf and Pyd antibodies shows a significant reduction of nephrocyte diaphragm length in larval garland nephrocytes expressing high levels (28°C) of Piezo wild type (Piezo WT), whereas cells expressing the mutant without the mechanosensitive channel activity (Piezo w/o MS) do not exhibit a phenotype. Green = Duf; magenta = Pyd. Scale bar = 5 µm. One-way ANOVA with Tukey's multiple comparisons test: *$P < 0.05$. **(C, D)** FITC-Albumin uptake assays do not reveal any phenotype in larval garland cells at 25°C (C), whereas overexpression of wild-type Piezo (Piezo WT) at 28°C results in a significantly increased FITC-Albumin uptake (D). Control: w; sns-Gal4/+;UAS-dicer2/+. Piezo WT: w; sns-Gal4/+;UAS-dicer2/UAS-Piezo-GFP. Piezo w/o MS: w; sns-Gal4/+;UAS-dicer2/UAS-Piezo-2306MYC.FLAG. t test: ***$P < 0.001$.

haemolymph clearance is surprising, given that the overexpression of Piezo results in elevated uptake of FITC-Albumin, potentially mediated via greater accessibility of Cubilin and increased Cubilin expression. Although one might expect a similar elevation in $AgNO_3$ uptake upon Piezo overexpression, because $AgNO_3$ can also be taken up by Cubilin (Zhang et al, 2013), additional effects of Piezo overexpression in nephrocytes might influence pupal development, as this is delayed already under basal conditions without feeding $AgNO_3$. The $AgNO_3$ toxin assay also differs from the FITC-Albumin uptake assay in that it can be considered a long-term read-out, assessing uptake function over days, whereas the FITC-Albumin uptake assay is a short-term read-out, focusing on uptake within minutes only. Also, it is possible that nephrocytes do filter/take up factors including $AgNO_3$ from the haemolymph with a higher rate upon overexpression of Piezo. However, this might also include the uptake of non-toxic or beneficial factors, which are needed in the haemolymph for normal development. Thus, differences in effects of FITC-Albumin uptake and clearance function/developmental delay might be observed here. As the whole-body knockout of Piezo most likely impacts on larval and pupal

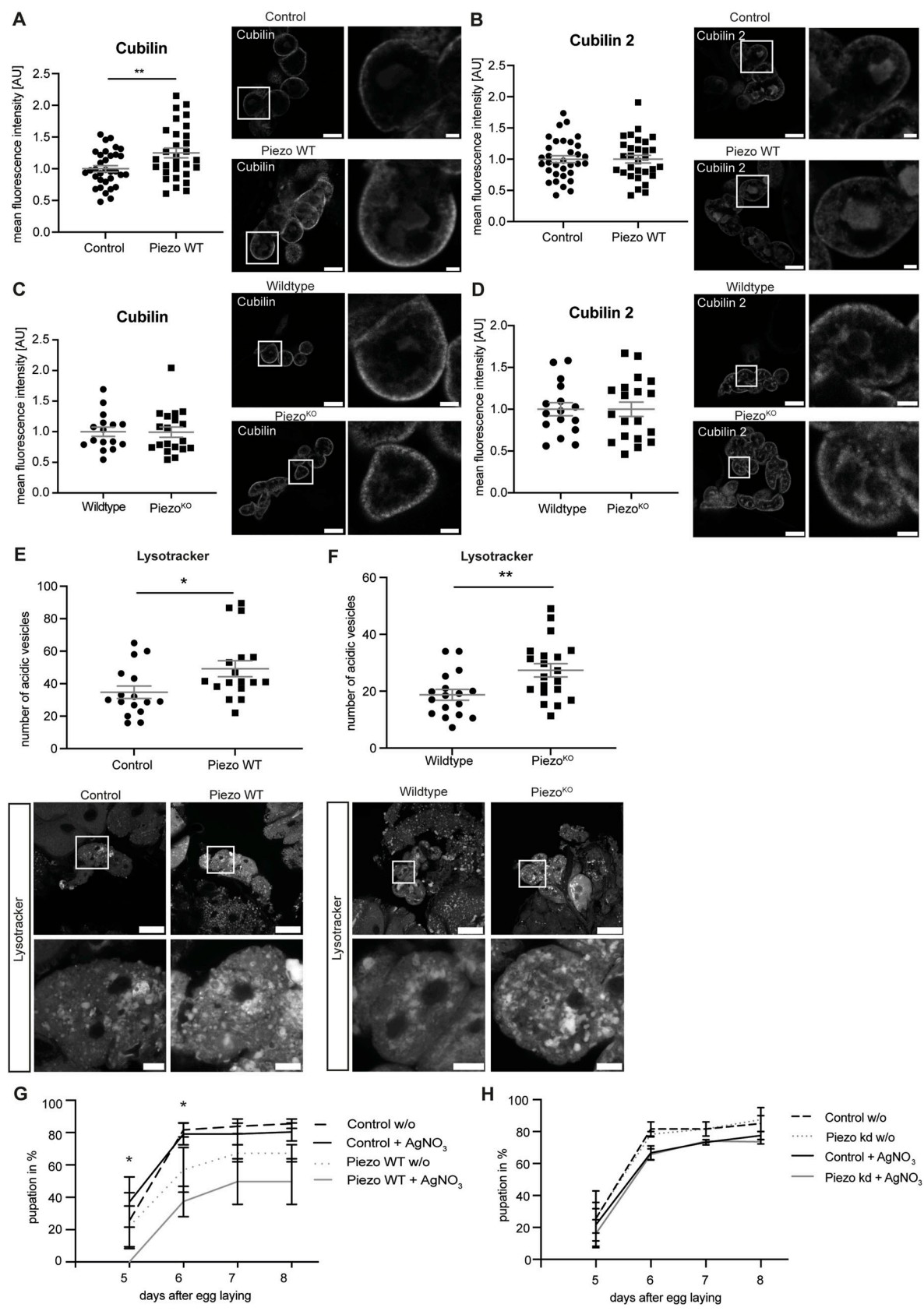

development independently of nephrocyte clearance function, we did not assess the Piezo[KO] in the AgNO$_3$ assay, but only the RNAi-mediated nephrocyte-specific knockdown. This revealed no significant differences in pupation without AgNO$_3$ (baseline) and no delay upon feeding the toxin (Fig 4H).

Taken together, the overexpression of Piezo wild type results in an increased expression of the receptor Cubilin—which might be more easily accessible because of the morphological alterations observed at the nephrocyte diaphragm, as well as more acidic vesicles and impaired haemolymph clearance.

### Piezo wild-type overexpression results in accumulation of actin fibres and an increase of superoxide, and seems to influence activity levels of the GTPase Rho1

As the overexpression of Piezo wild type results in a more severe phenotype then Piezo depletion, we focused only on additional downstream pathways and structures that are potentially influenced by elevated Piezo expression. We assessed actin fibres at the cell cortex and the cell surface in larval garland nephrocytes. Piezo is a cation channel that enables Ca$^{++}$ influx in nephrocytes (Schulz et al, 2024; Zhao et al, 2024); moreover, Ca$^{++}$ impacts on the actin cytoskeleton in general (Gasperini et al, 2017). Therefore, we examined actin fibre formation upon overexpression of Piezo wild type using phalloidin staining. Here, in nephrocytes overexpressing Piezo wild type at 28°C we observed a significant increase in actin fibres at the cell surface and cell cortex (Fig 5A and B).

Also, accumulation of actin fibres at the cell cortex in podocytes is partially mediated by increased RhoA-GTP levels (Wang et al, 2012). Moreover, Piezo has been previously reported to impact on GTPases (Canales Coutiño & Mayor, 2021; Zhang et al, 2021). Thus, we tested whether Piezo and Rho1 are linked in nephrocytes and whether the observed FITC-Albumin phenotype might be influenced by altered GTPase activation. Here, we focused on the GTPase Rho1. Because of the lack of a *Drosophila*-specific Rho1 antibody, we instead investigated the genetic interaction and assessed a potential rescue of the Piezo wild-type phenotype through simultaneous expression of a dominant-negative variant of Rho1 (Rho1-DN). This revealed a significantly lower FITC-Albumin uptake in Piezo wild type–overexpressing garland nephrocytes when Rho1-DN is simultaneously expressed, whereas the exclusive expression of Rho1-DN did not cause a phenotype (Fig 5C). To exclude dilution effects of the Gal4-UAS system, we tested the effect of simultaneous expression of Piezo wild type and GFP. Here,

we observed the expected increased FITC-Albumin uptake (Fig S4E). Of note, in this experiment we assessed FITC-Albumin uptake with an anti-albumin antibody labelled with Cy3 to avoid overlap with the GFP signal.

Mitochondria are known to be potent buffers of cytosolic Ca$^{++}$ through the storage of excess Ca$^{++}$. Because Piezo is a cation channel, overexpression might result in the intracellular accumulation of Ca$^{++}$ and thus impact on mitochondria (Giorgi et al, 2018). Increased Ca$^{++}$ levels in mitochondria have been reported to activate enzymes of the tricarboxylic acid cycle (TCA) such as the pyruvate dehydrogenase complex and oxidative phosphorylation, with further impacts on mitochondrial morphology also described (Gellerich et al, 2010; Deheshi et al, 2015; Görlach et al, 2015; Angelova & Abramov, 2024). Moreover, overactivation of mitochondrial enzymes and thus an increased metabolic rate are linked to the production of reactive oxygen species (ROS) in mitochondria (Starkov et al, 2004; Görlach et al, 2015). Hence, here we also investigated mitochondria and ROS formation upon overexpression of Piezo wild type. Using MitoTracker to visualize mass and morphology of mitochondria in addition to activity, we observed an increased signal intensity upon Piezo wild-type overexpression (Fig 5D). To further assess whether this elevation of signal intensity is related to mitochondrial activity, we also analysed ROS production in particular looking at superoxide. This demonstrated a significant increase of MitoSOX (mitochondrial superoxide indicator) intensity and consequently of superoxide also (Fig 5E).

Taken together, we observed an accumulation of actin stress fibres upon overexpression of Piezo, elevation of mitochondria mass and/or activity, and increased superoxide levels. Also, our data suggest a role of Rho1 downstream of Piezo, with a potential involvement in the observed FITC-Albumin phenotype. Of note, a recent study investigated GTPase activation in podocytes and observed less RhoA activation upon loss of Piezo (Melica et al, 2025). Our data highlight the importance of Piezo and its involvement in several important downstream signalling pathways in larval garland nephrocytes, which might result in pathological consequences.

### Short-term pharmacological inhibition of Piezo with the tarantula toxin GsMTx4 reverses the nephrocyte phenotypes

As overexpression of Piezo wild type induced both morphological and functional phenotypes and impacted on multiple downstream

---

**Figure 4. Overexpression of Piezo results in increased Cubilin levels, more acidic vesicles, and less haemolymph clearance.**
**(A)** Antibody-based staining of the receptor Cubilin shows a significant fluorescence increase upon overexpression of Piezo wild type. Scale bar = 25 and 5 $\mu$m (magnification). $t$ test: **$P < 0.01$. Control: w; sns-Gal4/+;UAS-dicer2/+. Piezo WT: w; sns-Gal4/+;UAS-dicer2/UAS-Piezo-FLAG. **(B)** Staining with an anti-Cubilin 2 antibody does not reveal any changes of receptor expression upon Piezo wild-type overexpression. Scale bar = 25 and 5 $\mu$m (magnification). **(C)** Visualization of Cubilin in Piezo-depleted garland nephrocytes does not show any alterations. Scale bar = 25 and 5 $\mu$m (magnification). Piezo[KO]: w;+;Piezo[ko]; Wildtype: w[1118]. **(D)** Cubilin 2 visualization also does not reveal any changes upon depletion of Piezo. Scale bar = 25 and 5 $\mu$m (magnification). **(E)** LysoTracker staining shows a significant increase of acidic vesicles in Piezo wild type–overexpressing nephrocytes at 28°C. Scale bar = 25 and 5 $\mu$m (magnification). $t$ test: *$P < 0.05$. **(F)** Piezo-depleted nephrocytes present with more acidic vesicles based on LysoTracker staining. Scale bar = 25 and 5 $\mu$m (magnification). $t$ test: **$P < 0.01$. **(G)** AgNO$_3$ toxin assays confirm a severe filtration and haemolymph clearance defect, as larvae expressing Piezo wild type in nephrocytes present with a significantly delayed pupation behaviour already at 25°C. Control: w; sns-Gal4/+;UAS-dicer2/+. Piezo WT: w; sns-Gal4/+;UAS-dicer2/UAS-Piezo-GFP. Two-way ANOVA with Tukey's multiple comparisons test: *$P < 0.05$. Depicted are means ± SEM. Each timepoint represents three independent n's with 15–20 first instar larvae each. **(H)** AgNO$_3$ toxin assay using Piezo RNAi does not reveal any differences in pupation behaviour, when compared to control flies. Control: w;sns-Gal4/+;UAS-dicer2/+; Piezo kd: w;sns-Gal4/UAS-Piezo-RNAi;UAS-dicer2/+. Depicted are means ± SEM. Each timepoint represents three independent n's with 15–20 first instar larvae each.

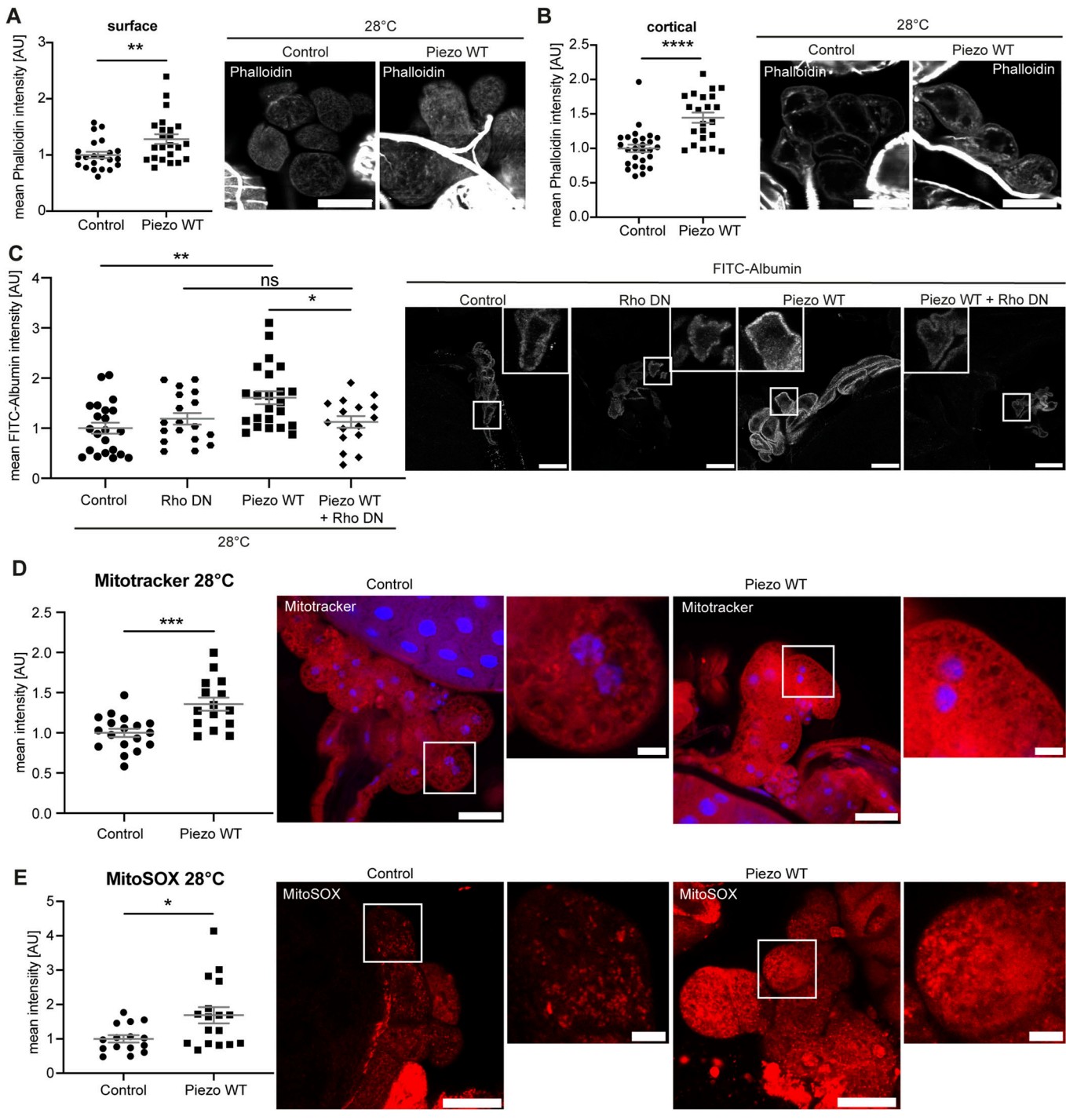

**Figure 5. Piezo wild-type overexpression results in accumulation of actin fibres in garland nephrocytes.**
**(A, B)** Visualization of actin fibres with phalloidin reveals a significant increase of surface (A) and cortical (B) actin stress fibres in larval garland nephrocytes expressing Piezo wild type (Piezo WT) at 28°C. t test: **$P < 0.01$; ****$P < 0.0001$. Scale bar = 25 μm. Control: w; sns-Gal4/+;UAS-dicer2/+. Piezo WT: w; sns-Gal4/+;UAS-dicer2/ UAS-Piezo-FLAG. **(C)** Simultaneous expression of Piezo wild type (Piezo WT) with dominant-negative Rho1 (Rho-DN) at 28°C results in a significant decrease in FITC-Albumin uptake when compared to Piezo wild-type overexpression (Piezo WT). Expression of Rho-DN alone does not impact on FITC-Albumin uptake. Scale bar = 25 μm. One-way ANOVA with Tukey's multiple comparisons test: *$P < 0.05$; **$P < 0.01$. Control: w; sns-Gal4/+;UAS-dicer2/+; Rho-DN: w; sns-Gal4/+;UAS-dicer2/ UAS.rho1.N19; Piezo WT: w; sns-Gal4/+; UAS-Piezo-GFP/+; Piezo WT + Rho-DN: w; sns-Gal4/+; UAS.rho1.N19/UAS-Piezo-GFP. **(D)** MitoTracker staining reveals a significant increase of fluorescence intensity in garland nephrocytes overexpressing Piezo wild type at 28°C. t test: ***$P < 0.001$. Scale bar = 25 and 5 μm (magnification). Control: w; sns-Gal4/+;UAS-dicer2/+. Piezo WT: w; sns-Gal4/+;UAS-dicer2/UAS-Piezo-FLAG. **(E)** MitoSOX stainings show a significant increase of superoxide in nephrocytes isolated from flies overexpressing Piezo wild type. t test: *$P < 0.05$. Scale bar = 25 and 5 μm (magnification).

pathways, we next investigated whether the tarantula toxin GsMTx4, a non-specific Piezo channel inhibitor, could alleviate these phenotypes. First, we assessed nephrocyte morphology and demonstrated that treatment of Piezo wild type–overexpressing garland nephrocytes with GsMTx4 results in a morphology similar to control cells (Fig 6A and B). Moreover, GsMTx4 treatment reversed the elevated FITC-Albumin uptake upon Piezo overexpression, further supporting our hypothesis that GsMTx4 might have a beneficial effect (Fig 6C and D). As GsMTx4 treatment exhibited a beneficial, but not significant, effect in the setting of a short-term treatment (5 min), we next assessed whether long-term treatment could also be beneficial in preventing the phenotypes associated with elevated Piezo levels. We therefore used a previously described fly strain expressing the active peptide of GsMTx4 (GsMTx4-AP) under the control of UAS (Beqja et al, 2020), which is expressed from embryonic stage 12–14 in our case (*sns*-Gal4). Although overexpression of GsMTx4 did not impact on garland nephrocyte morphology, FITC-Albumin uptake was significantly decreased (Fig S5A and B). This finding shows that the expression of GsMTx4-AP starting in embryonic stages causes a pathological phenotype in larval garland nephrocytes. As GsMTx4 is not specific for Piezo, this detrimental effect on nephrocyte function could be mediated by the inhibition of several channels important for nephrocyte biology.

Taken together, a short-term treatment with GsMTx4 reversed the nephrocyte phenotype upon Piezo overexpression, but long-term exposure caused pathological effects itself, showing the need to carefully assess GsMTx4 dosage and treatment duration in future experiments involving its use as a novel intervention.

# Discussion

Mechanosensation is an important signalling mechanism in podocytes, as these cells are constantly exposed to biomechanical forces (Kriz & Lemley, 2015). Such forces increase during diabetic nephropathy and hypertension, and elevated fluid shear stress can cause podocyte loss (Kriz & Lemley, 2015; Puelles et al, 2016; Haley et al, 2018). Interestingly, a putative adaptive and protective mechanism in podocytes has been described in response to elevated biomechanical force. In this mechanism, podocytes exhibit a hypertrophy phenotype to cover blank capillaries resulting from the loss of neighbouring cells (Wiggins et al, 2005; Kriz & Lemley, 2015, 2017). How podocytes sense and respond to alterations in biomechanical force—and subsequently induce compensatory mechanisms such as hypertrophy—is not understood in detail. However, we and others have previously identified and characterized mechanosensors and mechanotransducers in podocytes, among them, TRP6C, YAP, Talin, and Filamins (Winn et al, 2005; Möller et al, 2007; Tian et al, 2014; Rinschen et al, 2017; Koehler et al, 2020, 2022; Greiten et al, 2021). Another mechanotransducer that has recently gained interest in the renal community is Piezo. In mammals, two Piezo channels are expressed: Piezo1, which is involved in sensing blood flow–associated shear stress (Cahalan et al, 2015; Retailleau et al, 2015); and Piezo2, which responds to proprioception and touch (Zhong et al, 2018). In *Drosophila*, there is

only one Piezo variant, which we and others have previously described in adult pericardial nephrocytes (Schulz et al, 2024; Zhao et al, 2024). In brief, loss and overexpression of Piezo resulted in a nephrocyte phenotype, including morphological and functional disturbances, reflecting its critical role in this cell type. Within this study, we focused our research on larval garland nephrocytes. In contrast to adult pericardial nephrocytes, which lie along the heart tube and thus are exposed to biomechanical forces induced by heart movement, garland nephrocytes are positioned around the oesophagus. These cells are also exposed to biomechanical forces because of haemolymph movement within the open circulatory system, motion of the oesophagus while eating, and movement of the larvae itself. To assess whether garland nephrocytes can detect and respond to biomechanical forces, we applied a mechanical stimulus ex vivo and investigated their FITC-Albumin uptake. Both nephrocyte cell types present with significantly reduced uptake of FITC-Albumin into the lacuna system and thus seem to express protein machinery able to sense and respond to these forces. Based on these findings, we then investigated the effect of Piezo depletion and overexpression in larval garland cells. By doing so, we could show a mild functional phenotype upon loss of Piezo, whereas the overexpression of the channel resulted in a more severe phenotype including morphological and functional alterations. Of note, in *Drosophila* wing discs, similar phenotypes are observed upon both overexpression and depletion of Piezo, although these phenotypes presumably result from different underlying mechanisms (Mim et al, 2024). Interestingly, depletion of Piezo in nephrocytes and simultaneous application of a mechanical stimulus resulted in the absence of a significant FITC-Albumin uptake phenotype, suggesting an involvement of Piezo in this phenotype. However, future studies should to be done to examine the role of other mechanotransducers and downstream signalling pathways potentially contributing to this phenotype as well.

In our study, we observed an increased FITC-Albumin uptake upon depletion or overexpression of Piezo wild type. To further delineate involved mechanisms in FITC-Albumin uptake, we assessed the expression of the receptors Cubilin and Cubilin 2, as well as the number of acidic vesicles. Although Piezo depletion revealed only an increase of acidic vesicles, the overexpression of Piezo wild type also resulted in increased Cubilin levels, as well as a significant increase of acidic vesicles. Cubilin is involved in ligand internalization, such as albumin in mammalian proximal tubule cells; this function is likely conserved in *Drosophila* nephrocytes. The elevated FITC-Albumin uptake, increased Cubilin levels, altered nephrocyte diaphragm morphology, and increased number of acidic vesicles in Piezo wild type–overexpressing nephrocytes observed in this study are in line with this hypothesis. Whether Cubilin levels are elevated as a direct result of Piezo overexpression or whether accessibility of Cubilin is improved because of morphological alterations remains unknown. The elevated FITC-Albumin uptake in Piezo[KO] nephrocytes, however, does not seem to be mediated through elevated Cubilin levels. The increased FITC-Albumin uptake might be a result of side effects induced by the whole-body knockout, as nephrocyte-specific Piezo depletion with RNAi did not result in significant changes of FITC-Albumin uptake and haemolymph clearance is not affected by loss of Piezo. In

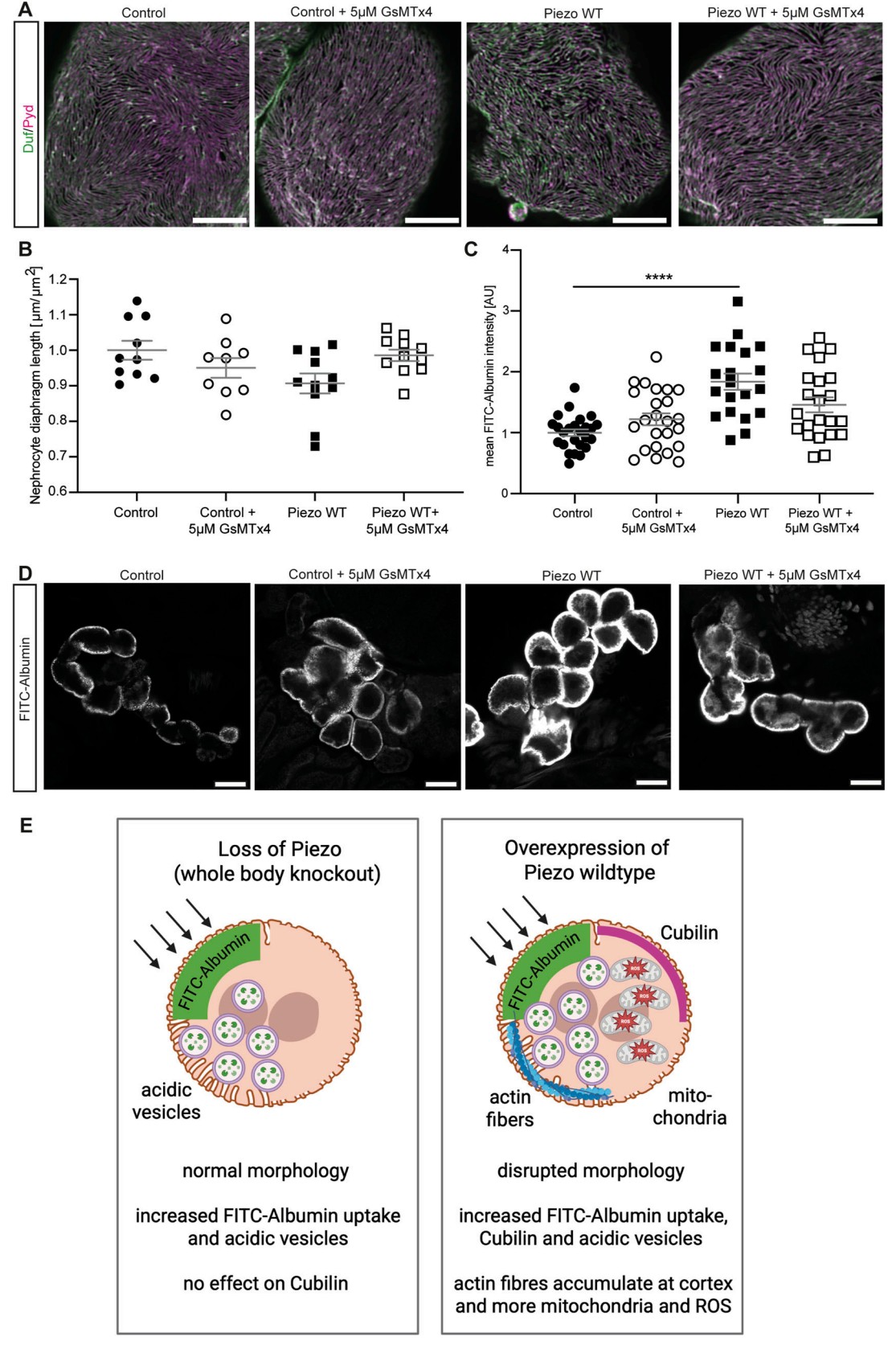

addition, we cannot exclude the involvement of other components of the nephrocyte uptake machinery such as Amnionless, which was not assessed within this project. Other downstream pathways including endocytosis and degradation processes might also be altered upon loss of Piezo, impacting on FITC-Albumin processing once it has been taken up.

The impact of Piezo on acidic vesicles and autophagy has been previously reported by Zhao et al (2024). They used adult pericardial nephrocytes depleted of Piezo and described an increase in Ref(2)p aggregates, which is considered as a marker of impaired autophagic flux (Zhao et al, 2024). In line with this, loss of Piezo resulted in an accumulation of non-acidic autophagosomes (Atg8-positive vesicles) and lower numbers of acidic vesicles (LysoTracker-positive vesicles), confirming the role of Piezo in maintaining normal autophagic flux (Zhao et al, 2024). Of note, nephrocytes have been shown to strongly depend on mechanisms such as autophagy, endocytic membrane trafficking/endocytosis, and organelle acidification to maintain physiological conditions (Fu et al, 2017; Wang et al, 2021; Lang et al, 2022; Sivakumar et al, 2022; Zhu et al, 2023).

The actin cytoskeleton plays an important role in podocyte biology and is essential for rearrangement processes (Wang et al, 2020). Mechanosensors and Mechanotransducers, in particular cation channels, might impact on the actin cytoskeleton through changes in intracellular Ca$^{++}$ levels (Lehne & Bogdan, 2023). In a previous study, nephrocytes were investigated in regard to the actin cytoskeleton and presented with actin clusters and cortical actin under physiological conditions (Muraleedharan et al, 2018). Moreover, we have demonstrated before that elevated Piezo wild-type levels cause a significant increase of cortical and surface actin fibres in adult pericardial cells (Schulz et al, 2024). Thus, within this study, we also investigated the impact of Piezo overexpression on the actin cytoskeleton in larval garland cells. In doing so, we corroborated our findings from adult pericardial cells, as elevated Piezo wild-type levels caused significantly higher levels of actin fibres at the cortex and surface of larval garland nephrocytes. Melica et al (2025) recently investigated the involvement of Piezo in actin fibre formation in cultured podocytes. They demonstrated a reduction of actin fibres upon loss of Piezo and an increase in stress fibre formation after applying the Piezo1 activator Yoda1 (Melica et al, 2025). These findings support our results observed in *Drosophila* nephrocytes.

Given that altered intracellular Ca$^{++}$ levels also impact on GTPases, which can in turn regulate the actin cytoskeleton (Greka & Mundel, 2012), we also used a genetic approach to test whether the GTPase RhoA is involved in the observed Piezo overexpression phenotype. Of note, in cultured podocytes, it has been shown that the expression of constitutively active RhoA results in an increase

of actin fibres (Wang et al, 2012) and that the activity levels of small GTPases need to be tightly regulated to prevent podocyte injury (Zhu et al, 2011; Yu et al, 2013; Robins et al, 2017). Here, we intended to reverse the observed Piezo overexpression phenotype in the FITC-Albumin assay by simultaneously expressing a dominant-negative variant of Rho1 (RhoA) to inhibit this signalling mechanism downstream of Piezo. Indeed, the increased FITC-Albumin uptake was prevented by expressing the dominant-negative Rho1 together with Piezo wild type. However, additional unspecific effects on endocytosis or ER stress through dominant-negative Rho1 expression cannot be excluded. Interestingly, Piezo itself is also known to impact on the ER, as loss of the channel results in expansion of the ER and increased ER stress in adult pericardial nephrocytes (Zhao et al, 2024).

In line with the beneficial effect of inhibiting Rho1 activity, we have previously shown a severe morphological and functional nephrocyte phenotype upon expression of constitutively active Rho1 (Koehler et al, 2021). Similar to our data, which suggests a link between Piezo and GTPases, three studies using mammalian models have shown regulation of GTPases through Piezo1. Loss of Piezo1 was associated with decreased levels of active RhoA and Rac1, whereas activation of Piezo1 resulted in an increase of active Rac1 levels (Fu et al, 2024; Ogino et al, 2024; Melica et al, 2025). Moreover, interfering with GTPase activation also impacted on actin cytoskeleton remodelling, consistent with our results investigating nephrocytes overexpressing Piezo wild type.

Because mitochondria act as major Ca$^{++}$ buffers (Giorgi et al, 2018) and Piezo overexpression may lead to excessive uptake of Ca$^{++}$ and increased intracellular Ca$^{++}$ levels, we assessed mitochondrial morphology and activity. This analysis revealed an increase in mitochondrial mass and/or activity, as well as elevated superoxide levels in nephrocytes overexpressing Piezo. In line with our findings, Li et al also reported that Piezo1 stimulates mitochondrial function (Li et al, 2022). Moreover, it has been shown that mechanical activation of Piezo impacts on ROS signalling in cardiomyocytes (Jiang et al, 2021).

In addition to the established links between Piezo, Ca$^{++}$, mitochondria, and ROS, the latter three have been described to play a role in nephrocyte biology. The presence and relevance of mitochondria in nephrocyte biology have been previously reported by Zhu et al (2024). In detail, disruption of mitochondrial dynamics by interfering with the Pink1–Park pathway results in morphological and functional disturbances, impaired endocytic membrane trafficking, and increased ROS levels (Zhu et al, 2024). Further studies describe an impact of ROS on p38 MAPK signalling in pericardial nephrocytes (Lim et al, 2014, 2019) and have shown oxidative stress as a consequence of defective endocytosis and also as a mediator of slit diaphragm defects (Xi et al, 2024). Ca$^{++}$

**Figure 6.  Pharmacological inhibition of Piezo with GsMTx4 reverses the nephrocyte phenotypes.**
**(A, B)** Immunofluorescence staining with Duf and Pyd antibodies reveals at least a partial rescue of the morphological changes in larval garland nephrocytes expressing Piezo wild type after incubation with 5 μM GsMTx4 for 5 min. Scale bar = 5 μm. Control: w; *sns*-Gal4/+;UAS-*dicer2*/+. Piezo WT: w; *sns*-Gal4/+;UAS-*dicer2*/UAS-*Piezo*-FLAG. **(C, D)** FITC-Albumin uptake assays show a rescue of the FITC-Albumin uptake phenotype observed in larval garland nephrocytes overexpressing Piezo wild type (Piezo WT) after 5 min of GsMTx4 treatment. One-way ANOVA with Tukey's multiple comparisons test: ****$P < 0.0001$. Scale bar = 25 μm. **(E)** Cartoon highlights the effect of Piezo depletion and overexpression. Although loss of Piezo only results in a mild phenotype with an increased FITC-Albumin uptake and acidic vesicles, the overexpression of Piezo causes morphological and functional disturbances. Moreover, Cubilin expression is elevated and more acidic vesicles are observed. Putative downstream pathways such as actin stress fibre formation, GTPase activation, and mitochondria and ROS formation are impacted as well. Images have been created with Biorender.com.

signalling has been described not only in the context of Piezo within pericardial nephrocytes (Schulz et al, 2024; Zhao et al, 2024), but also via store-operated calcium entry, highlighting the broader importance of Ca$^{++}$ signalling in nephrocytes (Sivakumar et al, 2022).

Based on our previously published findings of increased Piezo1 levels during chronic kidney disease in podocytes and the findings of other groups showing elevated Piezo1 levels in injury models, we tested the effect of a treatment—tarantula toxin (GsMTx4) on larval garland nephrocytes overexpressing Piezo wild type, similar to our previous approach in adult pericardial nephrocytes (Schulz et al, 2024). Our data confirm the beneficial effect of GsMTx4 on Piezo overexpression––associated phenotypes when applied short term, although effects are mild. This beneficial effect has also been described in murine models, in which treatment with GsMTx4 improved kidney function in a lupus nephritis mouse model and in the tubulointerstitial fibrosis in mice with unilateral ureter obstruction (UUO) or with folic acid treatment (Zhao et al, 2022). These findings show the potential of GsMTx4 as a novel therapeutic treatment in Piezo-associated kidney disease. Long-term application of GsMTx4 in nephrocytes using a genetic approach, however, resulted in pathological effects, highlighting the need to assess GsMTx4 treatment conditions and duration in detail.

Taken together, our findings show that both larval and pericardial nephrocytes respond to biomechanical stimuli. We show that Piezo is expressed in garland nephrocytes and localizes to the cell surface and cortical region close to the nephrocyte diaphragm. Applying a mechanical stimulus to nephrocytes depleted of Piezo revealed no significant FITC-Albumin uptake phenotype, suggesting an involvement of Piezo in the development of this phenotype induced by a mechanical stimulus. Future studies should investigate whether additional cation channels and mechanotransducers are involved and how this effect is mediated. Moreover, the overexpression of Piezo wild type results in a severe garland nephrocyte phenotype, characterized by morphological and functional disturbances, increased Cubilin expression, and a higher number of acidic vesicles (Fig 6E). Additional effects on the actin cytoskeleton and mitochondria, as well as the involvement of the GTPase Rho1, were observed in nephrocytes overexpressing Piezo wild type (Fig 6E). Based on these findings, we conclude that the increase of Piezo levels and the potential increase of intracellular Ca$^{++}$ influencing several downstream pathways and structures are detrimental for larval garland nephrocytes (Fig 6E). This hypothesis is in line with recently published studies, which investigated the expression of Piezo1 in podocytes under physiological conditions and during injury. All studies confirm the expression of Piezo1 in murine and human podocytes (Fu et al, 2024; Ogino et al, 2024; Schulz et al, 2024). Moreover, Piezo1 expression is elevated in podocytes exposed to shear stress and in murine models of hypertensive nephropathy, diabetic nephropathy, and human and mouse lupus nephritis, suggesting an important role of the mechanosensor during injury (Fu et al, 2024; Ogino et al, 2024; Li et al, 2025; Melica et al, 2025). Depletion of Piezo1 in podocytes resulted in protective effects in the lupus nephritis and diabetic nephropathy mouse model, providing further evidence of the pathological effect of elevated Piezo1 protein levels (Fu et al, 2024;

Li et al, 2025). Future studies should thus focus on further understanding downstream mechanisms mediated by Piezo and additional intervention options.

## Materials and Methods

### Fly husbandry

Flies were kept on standard fly food (Dundee recipe: protocol for 5 litres: agar: 42.6 g; brewer's yeast: 177 g; glucose: 314 g; maize: 285.6 g; yeast: 23 g; Nipagin: 10.6 g; ethanol: 83.3 ml; propionic acid: 13 ml; H$_2$O: 5 litres) at 25°C or 28°C. The UAS-Gal4 system is temperature-sensitive and causes increased expression of Gal4 and thus elevated protein expression at higher temperatures. Nephrocyte-specific expression of proteins or RNAi was achieved using the sns-Gal4 strain. All used fly strains are listed in Table 1.

### GsMTx4 treatment

Inhibition of Piezo was achieved by incubating nephrocytes with tarantula toxin/GsMTx4, an unspecific cationic mechanosensitive channel inhibitor. In detail, garland cells attached to the proventriculus were isolated and incubated with tarantula toxin/GsMTx4 for 5 min (5 $\mu$M in H$_2$O; control: HL3.1 buffer) (Cat. nr: STG-100; Alomone Labs).

### Immunofluorescence of *Drosophila* tissue

Garland nephrocytes attached to the proventriculus were isolated from third instar larvae. Isolation was performed in HL3.1 buffer (haemolymph-like). After isolation, nephrocytes were fixed in 4% formaldehyde for 20 min and 1 h in methanol, followed by three washing steps in 0.1% PBS/Triton X. Primary antibody incubation was done overnight at 4°C, and secondary antibody incubation has been done at room temperature for 45 min on day 2 (antibodies are listed in Table 2). Tissue was then mounted with ProLong Gold antifade mounting medium, before imaging. Quantification of nephrocyte diaphragm density was done using a FIJI macro (Koehler et al, 2021).

### FITC-Albumin uptake assay

To assess nephrocyte diaphragm integrity, FITC-Albumin uptake assays were performed as previously published (Hermle et al, 2017; Koehler et al, 2020). In detail, 0.2 mg/ml FITC-Albumin (in HL3.1) was incubated for 1 min, before washing with HL3.1 buffer for 1 min. Afterwards, samples have been fixed for 20 min in 4% formaldehyde. For UAS-GFP/UAS-Piezo-GFP–expressing nephrocytes, an anti-albumin antibody was used for visualization and quantification.

### AgNO$_3$ toxin assay

Flies of the appropriate genotype were allowed to lay eggs for 24 h at 25°C on juice plates with normal yeast. Afterwards, plates

**Table 1.  List of fly strains.**

| Fly strain | Origin | Purpose | Chromosome |
|---|---|---|---|
| Piezo MI{MIC} | BDSC ID: 60209 | Endogenous GFP-tag | 2. Chr. |
| Piezo[ko] | BDSC ID: 58770 | Knockout (backcrossed in w[1118] for six generations) | 2. Chr. |
| UAS-*Piezo*-FLAG | BDSC ID: 95296 | Overexpression | 3. Chr. |
| UAS-*Piezo*.2306MYC.FLAG | BDSC ID: 95298 | Overexpression | 3. Chr. |
| UAS-*Piezo*-GFP | BDSC ID: 58773 | Overexpression | 3. Chr. |
| UAS-*Piezo*-RNAi | VDRC ID: 105132 | Knockdown | 2. Chr. |
| UAS-GFP | BDSC ID: 5430 | Overexpression | 3. Chr. |
| sns-Gal4; UAS-dicer2 | The sns-Gal4 strain was obtained from S. Abmayr (Kocherlakota et al, 2008) and combined with UAS-dicer2 (Koehler et al, 2020) | Nephrocyte-specific driver | 2. + 3. Chr. |
| UAS-Rho1.N19 | BDSC ID: 7328 | Dominant-negative Rho1 | 3. Chr. |
| UAS-GsMTx4-AP (active peptide) | Strain obtained from B. Denholm (Beqja et al, 2020) | Overexpression | 3. Chr. |
| Oregon R | | Wild type | |
| W[1118] | | Wild type | |

**Table 2.  List of antibodies.**

| Name | Company/Provider | Catalogue nr./Reference | Host species | Dilution IF |
|---|---|---|---|---|
| Albumin | Abcam | ab207327 | Rabbit | 1:25 |
| Cubilin | M. Ruiz-Gomez | Atienza-Manuel et al (2021) | Rat | 1:200 |
| Cubilin 2 | M. Ruiz-Gomez | Atienza-Manuel et al (2021) | Guinea pig | 1:200 |
| Duf | M. Ruiz-Gomez | Weavers et al (2009) | Rabbit | 1:100 |
| GFP | Abcam | ab5450 | Goat | 1:100 |
| FLAG | Abcam | ab1162 | Rabbit | 1:100 |
| Pyd | Developmental Studies Hybridoma Bank | PYD2 | Mouse | 1:25 |
| Rab7 | Developmental Studies Hybridoma Bank | Rab7 | Mouse | 1:25 |

with eggs were kept at 18°C for 24 h. First instar larvae were then transferred to juice plates supplemented with yeast paste containing AgNO$_3$ (2 g yeast in 3.5 ml 0.003% AgNO$_3$). Plates were kept at 25°C for 4 d to let larvae develop and pupate. 5 d after egg laying, pupae were counted daily for four consecutive days. The number of pupae was normalized to the number of first instar larvae transferred to the plates.

### Application of a pressure stimulus

To apply a mechanical stimulus, the pressure injector SYS-PV830 (W.P.I. Instruments) was used. A 1-ml syringe fitted with a 26G needle as applicator nozzle was filled with HL3.1 buffer and positioned with the same angle and position relative to the bath solution's surface throughout the experiments. Isolated larval garland cells or adult pericardial nephrocytes were placed in a glass dish, and the stimulus was applied by pressure-controlled release of the buffer (0.5 bar for 3 s). After the stimulus, cells were incubated for 5 min before further assessment such as FITC-Albumin uptake assays.

### FISH

Garland nephrocytes (attached to proventriculus) were dissected from third instar larvae in HL3.1 buffer. Samples were fixed in 4% formaldehyde followed by methanol. FISH was then performed using the HCR Gold RNA-FISH Kit (Molecular Instruments) following the protocol from Bruce et al (2021). The Piezo probe (with up to 20 binding sites) was designed by Molecular Instruments (https://store.molecularinstruments.com/new-kit/gold/rnafish) based on the infinite catalogue for use with the X3 amplifier.

Samples were then mounted in ProLong Gold antifade reagent before imaging at a LSM 800 confocal Plan-Apochromat 63x/1.40 Oil DIC M27 objective (N.A. 1.4) combined with Airyscan and a zoom factor of 0.5. Piezo-specific mRNA particles were counted using Fiji (version ImageJ 1.54f). Images have been duplicated and

processed by applying a Gaussian blur with sigma = 16 (Image 2) and 2 (Image 1), respectively, followed by using the "Image Calculator" function to subtract background. The threshold for every subtracted image was set using the autothreshold algorithm "Default." The "Analyze Particles" function was used to count all particles within nephrocytes.

### LysoTracker staining

LysoTracker Red DND-99 (10 $\mu M$; Invitrogen) was incubated for 5 min after dissection of nephrocytes. Afterwards, cells were fixed with 4% formaldehyde and mounted with ProLong Gold antifade mounting medium. Imaging was done using a Zeiss LSM 800 confocal microscope with a Plan-Apochromat 63x/1.40 Oil DIC M27 objective (N.A. 1.4) combined with Airyscan and a zoom factor of 0.7. For comparative analysis, the exposure time of experiments was kept identical.

Acidic vesicles (LysoTracker-positive) were counted using Fiji (version ImageJ 1.54 p). The images underwent preprocessing to enhance their suitability for analysis. In detail, images were converted to greys, duplicated, and processed by applying a Gaussian blur with sigma = 15 (Image 2) and 2 (Image 1), respectively. New images were created by subtracting these images using the "Image Calculator" function. Thresholds were determined using the autothreshold algorithm "Default" and the acidic vesicles counted using "Analyze Particles" with an area of at least 0.2 $(\mu m)^2$ to select all acidic vesicles larger than 500 nm in diameter.

### MitoTracker and MitoSOX treatment

MitoTracker Red CMXRos (20 nM in DMSO; Cell Signaling) was incubated for 15 min at 28°C followed by a fixation with ice-cold methanol on ice for 15 min. Afterwards, cells were washed three times with PBS before mounting. Isolated tissue was incubated with MitoSOX Red (2.5 $\mu M$ in DMSO; Invitrogen) for 30 min in the dark at 28°C. Afterwards, samples were washed with HL3.1 buffer, mounted, and imaged immediately. Imaging was done with a LSM 800 confocal from Zeiss, with a Plan-Apochromat 63x/1.40 Oil DIC M27 objective (N.A. 1.4) combined with Airyscan (only for Mito-Tracker) and a zoom factor of 1.1 (MitoSOX) and 1.3 (MitoTracker).

### Visualization and quantification of actin fibres

Visualization of actin fibres was performed by incubation with phalloidin 1:250 (Phalloidin-iFluor 647 conjugate; Biomol) for 45 min. Imaging was done with LSM 800 confocal from Zeiss, using a Plan-Apochromat 63x/1.40 Oil DIC M27 objective (N.A. 1.4) combined with Airyscan and a zoom factor of 1.1. For actin fibre quantification, cortical and surface mean intensities of phalloidin were measured. Quantification of cortical actin fibres was done using the brush selection tool from FIJI.

### Imaging and quantification

Imaging of garland nephrocytes was done using a LSM 5 confocal from Zeiss with a Plan-Neofluar 20x/0.5 Ph2 objective (N.A. 0.5) (only used for FITC-Albumin assay shown in Fig 2C), a Leica SP8 confocal with a HC PL APO CS2 20x/0.75 DRY objective (N.A. 0.75) (only used for FITC-Albumin assay shown in Fig 2D), and a Zeiss LSM 800 confocal microscope with a Plan-Apochromat 63x/1.40 Oil DIC M27 objective (N.A. 1.4) combined with Airyscan (only for antibody-based immunofluorescence; FITC-Albumin assays have been imaged without Airyscan). Images were further processed and analysed using ImageJ (version 1.53c). For comparative analysis, the exposure time of experiments was kept identical.

### Statistical analysis

Statistical significance was determined using GraphPad Prism software version 8 for Mac (GraphPad Software). Results are expressed as means ± SEM. One datapoint represents the mean value of all nephrocytes analysed from one fly from independent matings. Comparison of two groups was done using a $t$ test; comparison of more than two groups with one independent variable was done using one-way ANOVA followed by Tukey's multiple comparisons test. A $P$-value < 0.05 was considered statistically significant.

# Data Availability

All data needed to evaluate the conclusions in the article are present in the article or the Supplementary Materials.

# Supplementary Information

# Acknowledgements

We thank Philipp Antczak (University of Cologne) for helpful discussions and intellectual input. The Anti-Pyd monoclonal antibody developed by Fanning (Choi et al, 2011) was obtained from the Developmental Studies Hybridoma Bank, created by the NICHD of the NIH and maintained at the University of Iowa, Department of Biology, Iowa City, IA 52242. Images were created with BioRender.com. S Koehler received funding from the German Research Foundation (CRC1192/3, KO 6045/1-1 and KO 6045/4-1, SFB TRR 422/1 A04) and the Else Kröner Fresenius Foundation (2022_EKEA.09). TB Huber received funding from the German Research Foundation (CRC1192, HU 1016/8-2, HU 1016/11-1, HU 1016/12-1). S Koehler and TB Huber received funding from the BMBF (STOP-FSGS-01GM2202A).

### Author Contributions

P Hazelton-Cavill: data curation, formal analysis, investigation, methodology, and writing—review and editing.
KK Alornyo: data curation, investigation, methodology, and writing—review and editing.
M Bouchard: data curation, formal analysis, investigation, methodology, and writing—review and editing.
K Schulz: data curation and methodology.
TB Huber: funding acquisition and investigation.
B Denholm: investigation.

S Koehler: conceptualization, data curation, formal analysis, supervision, funding acquisition, investigation, visualization, methodology, project administration, and writing—original draft, review, and editing.

## Conflict of Interest Statement

The authors declare that they have no conflict of interest.

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
