## [Reviewer comments · Life Science Alliance]

Elevated Piezo levels cause structural and functional alterations in *Drosophila* garland nephrocytes

Paris Hazelton-Cavill, Karl Aloronyo, Michelle Bouchard, Kristina Schulz, Tobias Huber, Barry Denholm, and Sybille Koehler
DOI: <https://doi.org/10.26508/lsa.202503515>

Corresponding author(s): Sybille Koehler, Universität Hamburg

Review Timeline:

Submission Date:	2025-09-22
Editorial Decision:	2025-10-29
Revision Received:	2025-12-19
Editorial Decision:	2026-01-23
Revision Received:	2026-01-26
Accepted:	2026-01-27

Scientific Editor: Sarita Hebbar

Transaction Report:

October 29, 2025

Re: Life Science Alliance manuscript #LSA-2025-03515-T

Dr. Sybille Koehler
Universität Hamburg
Martinistr. 52
Hamburg 20257
Germany

Dear Dr. Koehler,

Thank you for submitting your manuscript entitled "Elevated Piezo levels cause *Drosophila* nephrocyte injury, impacting on actin fibres and mitochondria" to Life Science Alliance. Your manuscript was reviewed by three experts whose comments are appended below. As you will note, two of the three reviewers find the work interesting, timely, and of potential significance. That said, all the reviewers have raised significant concerns that preclude publication at this stage. We agree with the reviewers that the following concerns must be addressed/clarified in a revised manuscript:

1. Description of Piezo's expression and localisation must include the following clarifications: (a) its expression in normal physiology and if there is any specificity of garland cells expressing it (Reviewer 2) and (b) localisation close to nephrocyte diaphragm (Reviewer 3).
2. Data on efficiency of knockout of Piezo (Reviewers 1 and 2), using any of the suggested approaches, must be provided. We leave it optional for you to provide data using an additional RNAi line (as suggested by Rev 1, depending on availability). We agree with Reviewer 1 that you should also provide experimental evidence to back the statement on Piezo over-expression at higher temperatures.
3. Experiments involving FITC-Albumin uptake must also be better explained in terms of motivation for use of FITC-Albumin (Reviewer 1) and the discussion of its outcome (Reviewer 3). We encourage you to follow the point of Reviewer 1 to assay FITC-Albumin uptake with the combination of reduced Piezo levels and pressure stimulus.
4. Experiments involving LysoTracker staining must be better described (Reviewers 1 and 3). Further we also agree with Reviewer 1 that you must provide high resolution images and quantification to support the change in number of LysoTracker/acidic vesicles with changes in Piezo expression.
5. Expansion of the discussion as suggested by Reviewers 2 and 3. We also agree that you must tone down the conclusion of the study as stated in the discussion and abstract (Reviewers 1 and 3). We encourage you to follow the point of Reviewer 1 on modifying the title of the study.

In line with their overall assessment, we invite you to submit a revised manuscript addressing the reviewers' comments. When submitting the revision, please include a letter addressing the reviewers' comments point by point. While a rebuttal must respond to all points in some form, additional experiments to resolve these points, other than indicated above, will not be required. The typical timeframe for revisions is three months. Please note that papers are generally considered through only one revision cycle, so strong support from the referees on the revised version is needed for acceptance.

Thank you for this interesting contribution to Life Science Alliance. We are looking forward to receiving your revised manuscript.

Sincerely,

Sarita Hebbar, PhD
Scientific Editor
Life Science Alliance
<http://www.lsjournal.org>

B. MANUSCRIPT ORGANIZATION AND FORMATTING:

Reviewer #1 (Comments to the Authors (Required)):

The manuscript by Hazelton-Cavill et al. addresses the role of Piezo in *Drosophila* garland nephrocytes. The authors show that, by applying pressure, these nephrocytes are mechanosensitive in respect to protein uptake and express the mechanotransducer Piezo. Piezo knockout (in the whole animal) did not detectably alter morphology of garland cells, but did result in increased protein uptake (without applying pressure). Overexpression of Piezo altered nephrocyte morphology, increased protein uptake and resulted in multiple additional alterations, including increased Cubulin levels, more acidic vesicles, reduced hemolymph clearance and accumulation of F-actin fibers. Some of these alterations are reversed in the presence of GsMTx4, a non-specific Piezo channel inhibitor.

The manuscript reports novel data of the function of Piezo in garland nephrocytes. The conclusions that garland cells are mechanosensitive and express Piezo are well supported and are of interest. Whereas Piezo is required for this mechanosensation remains unclear. Moreover, the authors describe numerous alterations upon Piezo overexpression, which are of potential interest given that upregulation of Piezo is observed in some human diseases (e.g. Lupus nephritis). Mechanistic insights into the function of Piezo in nephrocytes are limited. The manuscript is well written, the data are clearly presented and conclusions are mostly appropriate.

Specific comments

1. Title: The use of the word 'injury' in the title does not seem to be justified. The authors describe molecular and morphological alterations and misfunctions of nephrocytes overexpressing Piezo, but, to the mind of this reviewer, not injury.
2. Fig. 1C,D. The authors should motivate the use of FITC -Albumin for the general reader.
3. Fig. 1C,D. The authors should indicate in the figure which markers are shown.
4. Fig. 1. Does the decrease of FITC-Albumin uptake in garland cells under external pressure depend on Piezo? The authors should knockdown Piezo in these cells (if efficient, see comment below), apply a pressure stimulus and measure FITC-Albumin uptake.
5. Fig. 2D. In contrast to a Piezo knockout, Piezo knockdown does not alter Albumin-FITC uptake. The authors should test the efficiency of their knockdown (using Piezo:GFP) and use a second, independent RNAi line to knockdown Piezo (if available).
6. Fig. 3. The authors claim that, due to the temperature-sensitivity of the Gal4 system, Piezo overexpression is stronger at 29C compared with 25C. Since the authors express a FLAG-tagged Piezo, the authors should directly test this assumption by anti-FLAG immunostaining.
7. Fig. 4E,F. The authors claim that Piezo depletion or overexpression results in a significant number of acidic vesicles. However, vesicles are difficult to detect in the images; moreover, the authors quantified mean intensities and not number of

vesicles positive for lysotracker. The authors should attempt to acquire higher resolution images and quantify number of lysotracker positive vesicles.

8. Line 274. The authors speculate that this increase of acidic vesicles might be a consequence of extensive FITC-Albumin uptake. However, to the understanding of this reviewer, no FITC-Albumin was used in these experiments.

9. Fig. 4G,H. The authors should mention what is shown in the plots (mean {plus minus}s.e.m?). Also, the authors should state the number of replicates analyzed.

10. Fig. 5A. The authors compare surface and cortical F-actin levels from control and Piezo overexpressing animals and report a significant difference. To test whether this difference is specific to F-actin, the authors should stain for a surface marker and a cortical marker (e.g. E-cadherin).

11. Line 493. The conclusion that "...our findings show [...] that Piezo plays an important role in larval garland nephrocyte..." does not seem to be justified given the "mild functional phenotype" (see Abstract) of Piezo depletion.

Reviewer #2 (Comments to the Authors (Required)):

This manuscript by Hazelton-Cavill et al. explores the role of the mechanotransducer Piezo in *Drosophila* nephrocytes as a model for mammalian podocytes. Using genetic manipulation (knockout, knockdown, and overexpression) and functional assays, the authors demonstrate that elevated Piezo levels cause nephrocyte injury associated with actin stress fiber accumulation, mitochondrial activation, oxidative stress, and increased Cubilin expression. They also show that short-term inhibition of Piezo with GsMTx4 partially rescues the phenotype. The study aims to contribute to understanding renal mechanotransduction and the pathological role of Piezo channels.

The topic of mechanosensitive ion channels in renal injury is timely and relevant. The use of *Drosophila* nephrocytes as a model system is well justified and aligns with previous work from the group. However, some points need further evaluation.

Major Comment

-Could the authors show the level of Piezo downregulation achieved with the RNAi construct in garland nephrocytes?

Demonstrating the level of knockdown, for example through quantitative PCR, immunostaining, or reporter signal reduction, would considerably strengthen the interpretation of the RNAi experiments.

-It would be helpful if the authors could indicate what proportion of garland nephrocytes normally express Piezo under physiological conditions. If expression is limited to a subset of cells, overexpression might induce Piezo in cells that are normally negative, potentially altering the phenotype.

-The authors might wish to discuss whether Piezo-positive nephrocytes could represent a distinct subpopulation with specific structural or functional properties, different from Piezo-negative cells. Addressing this point would provide a more nuanced understanding of the cell type-specific effects of Piezo signalling.

-Finally, I would encourage the authors to expand the discussion on the context- and cell type-specific roles of Piezo channels.

Minor Comment

-Colour for Duf in Figure1, panel E and F is unfortunate

-Fig 5 C, double bar in Piezo WT+ Rho DN

Reviewer #3 (Comments to the Authors (Required)):

Hazelton-Cavill and colleagues investigated the mechanosensitive ion channel Piezo in *Drosophila* garland cell nephrocytes. Exposing these cells to pressure reduced FITC-albumin endocytosis. Loss of Piezo did not affect the nephrocyte diaphragm but increased tracer uptake. High-level Piezo overexpression caused a mild reduction in nephrocyte diaphragms and also enhanced endocytosis of FITC-albumin and Cubilin, as well as Lysotracker labeling, actin, and mitochondrial markers.

The phenotypes observed are mostly mild or non-significant, and the investigation is somewhat lacking in structure and well-supported conclusions. The strong emphasis on excessive overexpression further weakens the work. Two previous studies, including one from the authors' own laboratory, have already examined Piezo function in pericardial nephrocytes. Overall, this study offers only rather modest novel biological insight.

Comments:

- I am not convinced of the physiological relevance of the pressure-stimulus experiments. Applying 0.5 bar (50,000 Pa) seems orders of magnitude above biologically plausible values. For comparison, shear stress in capillaries is around 0.1-5 Pa. Would a freely floating garland nephrocyte ever experience forces of this magnitude, or are the authors simply inducing mechanical cell damage?

- The Piezo::EGFP Mi{MIC} line was already characterized by Zhao et al., showing localization in channels. The present manuscript claims localization "close to the nephrocyte diaphragm" without providing adequately magnified cross-sections to support a conclusion divergent from Zhao et al.

- How is the increase in FITC-albumin uptake upon both loss and overexpression of Piezo explained, when only overexpression results in higher Cubilin levels?

- LysoTracker broadly labels different acidic compartments. It remains unclear what this indicates or how it relates to the phenotype. How is increased LysoTracker „a consequence of extensive FITC-albumin uptake“?
- Piezo overexpression increases endocytosis but diminishes toxin resistance. This contradictory finding remains unexplained. Does nephrocyte function improve or worsen?
- Higher mitochondrial abundance could follow a long list causes beyond cytosolic calcium, which was not even shown in this study.
- Why wasn't the combination of Rho DN and Piezo consistently controlled for GAL4 dilution within the same experiment?
- GsMTx4 hardly seems a plausible candidate for a "novel therapeutic treatment in Piezo associated kidney disease." The latter is not even an established clinical entity.
- The authors should more thoroughly discuss their findings in the context of previous studies on Piezo in nephrocytes (ER stress? Autophagy?).
- Some sections of the manuscript seem slightly wordy and at times repetitive.

Editorial comments:

We thank the editors for the careful consideration of our manuscript and addressed all comments and concerns as requested. Please find the detailed description below:

1. *Description of Piezo's expression and localisation must include the following clarifications: (a) its expression in normal physiology and if there is any specificity of garland cells expressing it (Reviewer 2) and (b) localisation close to nephrocyte diaphragm (Reviewer 3).*

Re a)

We have put together data from 8 Piezo-GFP MIMIC flies showing Piezo-GFP and Duf (nephrocyte marker) expression. We used Duf and GFP antibodies to visualize both proteins. Our data shows, that all cells positive for Duf also show GFP expression at the cell cortex. We thus concluded, that the majority of cells express Piezo under physiological conditions. Of note, we had to use an anti-GFP antibody to enhance the Piezo-GFP signal. This might be a result of low expression levels of Piezo under physiological conditions. Also, GFP levels differ between animals, which could be caused by antibody binding efficiency, as Duf levels also vary between flies. We decided to not include the data in the manuscript, but wanted to show our results here.

Also, by performing FISH with Piezo-specific mRNA probes in control flies (+; *sns*-Gal4/w;+) and wildtype (OreR) under physiological conditions (kept at 25°C, no additional treatment), we did not observe major differences between cells or between animals. All cells we analyzed did show Piezo-specific mRNA, the variation is considered biological variance between cells and animals. No control or wildtype animal had cells without any Piezo-specific mRNA. Again, we

did not include the data in the manuscript, but wanted to show our data here in the response letter.

However, we did add a sentence, stating that the majority of cells express Piezo, in the manuscript.

Re b):

We apologize if our conclusion was phrased in a misleading way, we do not propose a different localization from that described previously by Zhao et al.. We are really careful with the way in which we comment on localization based on immunofluorescence images, as we do not have any additional data, such as immunoprecipitation to claim Piezo expression at the nephrocyte diaphragm. To our understanding, Figure 1 in the Zhao et al manuscript shows UAS-Piezo-GFP in combination with the novel Sns-Ruby fly strain. The overexpression of Piezo results in expression in the lacunae, which we also observe in our overexpression flies as seen in Supplementary Figure 2A,C and E. We also observed a finger-print like pattern in the overexpression lines. In addition, Zhao et al see and describe in their manuscript: *'Using another Piezo-GFP reporter line [Piezo::EGFP Mi{MIC}], in which the endogenous Piezo is tagged by GFP, we confirmed the same cortical fingerprint pattern (Supplemental Figure 1, A and B).'* We used the same Piezo-GFP MIMIC line from BDSC and also observed a fingerprint like pattern. We have re-done imaging of Piezo-GFP MIMIC flies with a better resolution and to visualize the cortical area (Figure 1). We apologize, if our conclusion was not phrased clearly enough, but from our point of view, we have similar findings regarding Piezo localisation in wildtype (MIMIC)

and overexpression flies as to what has been reported for adult pericardial cells by Zhao et al. We thus rephrased our conclusion in the manuscript. Changes are marked in red.

2. Data on efficiency of knockout of *Piezo* (Reviewers 1 and 2), using any of the suggested approaches, must be provided. We leave it optional for you to provide data using an additional RNAi line (as suggested by Rev 1, depending on availability). We agree with Reviewer 1 that you should also provide experimental evidence to back the statement on *Piezo* over-expression at higher temperatures.

We performed additional experiments to confirm the knockdown efficiency of our *Piezo*-RNAi strain as suggested by reviewer 1 and 2. Of note, the same strain has been used by us and others before, to deplete *Piezo* in nephrocytes (1, 2). We could not perform Q-PCR, as we are not able to isolate nephrocytes and other tissue might also express *Piezo*, thus we might not be able to measure a decrease of *Piezo* in knockdown nephrocytes. Also, we could not generate stable flies expressing *Piezo*:GFP, *sns*-Gal4 and the RNAi strain due to time restrictions, to test reporter signal reduction. We do not have a *Piezo*-specific *Drosophila* antibody and thus decided against the immunostaining approach, but we recently established a fluorescence *in situ* hybridization (FISH) protocol in our lab and acquired *Piezo*-specific probes using the HCR™ Gold RNA-FISH tool from Molecular Instruments. By doing so, we have been able to visualize *Piezo*-specific mRNA in nephrocytes. Counting FISH signals (*Piezo*-specific mRNA) revealed a significant decrease in *Piezo* depleted nephrocytes when compared to controls. This data is now included in Supp. Figure 1A.

Based on these findings, we concluded that the knockdown is efficient and sufficient.

As suggested by reviewer 1, we assessed expression levels by staining for the FLAG tag. We indeed observed a significant increase of FLAG intensity from 25 to 28°C. We performed the experiments on the same day, using the same antibody and kept laser intensity equal during imaging. We also show a control, which has no FLAG staining. The data is now part of Supp. Figure 2. We also added a citation for the temperature-sensitivity of the Gal4 system (3).

3. Experiments involving FITC-Albumin uptake must also be better explained in terms of motivation for use of FITC-Albumin (Reviewer 1) and the discussion of its outcome (Reviewer 3). We encourage you to follow the point of Reviewer 1 to assay FITC-Albumin uptake with the combination of reduced Piezo levels and pressure stimulus.

We added an explanation for the general reader about why we choose the FITC-Albumin uptake assay:

The FITC-Albumin uptake assay is a widely used assay to assess nephrocyte function (4). FITC-Albumin can pass through the nephrocyte diaphragm and bind to receptors such as Cubilin and Amnionless, which are located in the lacunae membrane. After binding to these proteins, FITC-Albumin is endocytosed prior to degradation in vesicles including the lysosomes. In addition to detecting changes in nephrocyte uptake function, the FITC-Albumin assay can also indicate the occurrence of morphological alterations such as loss of nephrocyte diaphragm and reduced lacunae channels, which could result in a reduced number of receptors available for uptake (4). Thus, incubation of nephrocytes with FITC-Albumin can help identify alterations in endocytosis and nephrocyte diaphragm integrity as well as lacunae structure. Of note, both, decreased and increased FITC-Albumin levels, can point towards morphological alterations or changes in lysosomal degradation, as loss of nephrocyte diaphragms can also allow for easier access to receptors such as Cubilin (5).

We agree with reviewer 3, that the increased FITC-Albumin upon loss and overexpression of Piezo is an interesting finding, which we cannot explain at the moment. Our intention behind using the Cubilin antibody was to identify a possible mechanism explaining the increased FITC-Albumin uptake. However, the whole-body knockout of Piezo does not show alterations in Cubilin expression. In the discussion we added some additional thoughts in regard to this finding:

Additionally, we cannot exclude the involvement of other components of the nephrocyte uptake machinery such as Amnionless, which was not assessed within this project. Other downstream pathways including endocytosis and degradation processes might also be altered upon loss of Piezo, impacting on FITC-Albumin processing once it has been taken up.

In addition, we kept our previous explanation:

Of note, the elevated FITC-Albumin uptake in Piezo^{KO} nephrocytes does not seem to be mediated through elevated Cubilin levels. The increased FITC-Albumin uptake might be a result of

side effects induced by the whole-body knockout, as nephrocyte-specific Piezo depletion with RNAi did not result in significant changes of FITC-Albumin uptake and haemolymph clearance is not affected by loss of Piezo.

As suggested by reviewer 1, we performed additional experiments. Using our Piezo-RNAi at 25°C we assessed FITC-Albumin uptake in the absence and presence of a mechanical stimulus. While control cells respond with a decrease of FITC-Albumin uptake after applying a mechanical stimulus, Piezo knockdown cells do not show this phenotype and present with a partial rescue when compared to controls.

Supp. Fig. 1

Based on these findings we concluded, that the FITC-Albumin phenotype is at least partially mediated by Piezo. However, additional studies need to be done to examine the role of other mechanotransducers and downstream signaling pathways, as it is very unlikely Piezo is the only protein involved in this response. The new data is now included in Supp. Figure 1B.

4. Experiments involving LysoTracker staining must be better described (Reviewers 1 and 3). Further we also agree with Reviewer 1 that you must provide high resolution images and quantification to support the change in number of LysoTracker/acidic vesicles with changes in Piezo expression.

We agree with reviewer 3, lysotracker dyes accumulate in acidic organelles or vesicles and thus label lysosomes, late endosomes and other acidic vesicles. Thus, by using lysotracker we cannot distinguish between those vesicles. To further assess, which vesicles are altered upon depletion and overexpression of Piezo, we also analyzed Rab7, a late endosome marker. This did not reveal significant differences in nephrocytes expressing Piezo WT or in Piezo knockout cells. These findings suggest that changes observed in lysotracker positive vesicles are primarily not mediated by late endosomes. We kept the term 'acidic vesicles' in the manuscript, as we still cannot distinguish between lysosomes and other acidic vesicles, and as lysotracker has been previously used in *Drosophila* to identify organelle acidification (2, 6–8). The Rab7 data is now part of Supp. Figure 3.

We agree with reviewer 1 and 3, the wording of our conclusion was misleading. We changed this accordingly to:

This increase might be a result of extensive uptake of particles from the haemolymph – similar to the observed increase of FITC-Albumin uptake, which is subsequently degraded within nephrocytes.

As suggested by reviewer 1 we have re-done and re-imaged the lysotracker staining. By doing so we acquired higher resolution images and used FIJI to count vesicles positive for lysotracker and bigger than 500nm in diameter. The analysis as well as new representative images are now included in Figure 4E,F. We observed a significant increase of the number of vesicles positive for lysotracker. We kept our conclusion, as counting vesicles resulted in the same effect as measuring intensity and is in line with the increased FITC-Albumin uptake in both genotypes. We also added a description of the analysis in the material and methods section.

5. Expansion of the discussion as suggested by Reviewers 2 and 3. We also agree that you must tone down the conclusion of the study as stated in the discussion and abstract (Reviewers 1 and 3). We encourage you to follow the point of Reviewer 1 on modifying the title of the study.

We expanded the discussion as suggested by the reviewers and included additional thoughts, which arose during the revision process. We also rephrased the title and made it more descriptive taking out the term injury. All changes are marked in red.

Reviewer #1 (Comments to the Authors (Required)):

The manuscript by Hazelton-Cavill et al. addresses the role of Piezo in Drosophila garland nephrocytes. The authors show that, by applying pressure, these nephrocytes are mechanosensitive in respect to protein uptake and express the mechanotransducer Piezo. Piezo knockout (in the whole animal) did not detectably alter morphology of garland cells, but did result in increased protein uptake (without applying pressure). Overexpression of Piezo altered nephrocyte morphology, increased protein uptake and resulted in multiple additional alterations, including increased Cubulin levels, more acidic vesicles, reduced hemolymph clearance and accumulation of F-actin fibers. Some of these alterations are reversed in the presence of GsMTx4, a non-specific Piezo channel inhibitor.

The manuscript reports novel data of the function of Piezo in garland nephrocytes. The conclusions that garland cells are mechanosensitive and express Piezo are well supported and are of interest. Whereas Piezo is required for this mechanosensation remains unclear. Moreover, the authors describe numerous alterations upon Piezo overexpression, which are of potential interest given that up-regulation of Piezo is observed in some human diseases (e.g. Lupus nephritis). Mechanistic insights into the function of Piezo in nephrocytes are limited. The manuscript is well written, the data are clearly presented and conclusions are mostly appropriate.

We thank the reviewer for his/her comments and the valuable assessment of our manuscript.

Below we addressed all comments in detail:

Specific comments

1. Title: The use of the word 'injury' in the title does not seem to be justified. The authors describe molecular and morphological alterations and misfunctions of nephrocytes overexpressing Piezo, but, to the mind of this reviewer, not injury.

We rephrased the title and made it more descriptive taking out the term injury.

2. Fig. 1C,D. The authors should motivate the use of FITC -Albumin for the general reader.

We added an explanation for the general reader about why we choose the FITC-Albumin uptake assay:

The FITC-Albumin uptake assay is a widely used assay to assess nephrocyte function (4). FITC-Albumin can pass through the nephrocyte diaphragm and bind to receptors such as Cubilin and Amnionless, which are located in the lacunae membrane. After binding to these proteins, FITC-Albumin is endocytosed prior to degradation in vesicles including the lysosomes. In addition to detecting changes in nephrocyte uptake function, the FITC-Albumin assay can also indicate the occurrence of morphological alterations such as loss of nephrocyte diaphragm and reduced lacunae channels, which could result in a reduced number of receptors available for uptake (4). Thus, incubation of nephrocytes with FITC-Albumin can help identify alterations in endocytosis and nephrocyte diaphragm integrity as well as lacunae structure. Of note, both,

decreased and increased FITC-Albumin levels, can point towards morphological alterations or changes in lysosomal degradation, as loss of nephrocyte diaphragms can also allow for easier access to receptors such as Cubilin (5).

3. Fig. 1C,D. The authors should indicate in the figure which markers are shown.

We apologize for this and added the information in the images. All images show FITC-Albumin uptake and are representative images of the experiments.

4. Fig. 1. Does the decrease of FITC-Albumin uptake in garland cells under external pressure depend on Piezo? The authors should knockdown Piezo in these cells (if efficient, see comment below), apply a pressure stimulus and measure FITC-Albumin uptake.

We performed additional experiments as suggested. Using our Piezo-RNAi at 25°C we assessed FITC-Albumin uptake in the absence and presence of a mechanical stimulus. While control cells respond with a decrease of FITC-Albumin uptake after applying a mechanical stimulus, Piezo knockdown cells do not show this phenotype and present with a partial rescue when compared to controls.

Supp. Fig. 1

Based on these findings we concluded, that the FITC-Albumin phenotype is at least partially mediated by Piezo. However, additional studies need to be done to examine the role of other mechanotransducers and downstream signaling pathways, as it is very unlikely Piezo is the only protein involved in this response. The new data is now included in Supp. Figure 1B.

5. Fig. 2D. In contrast to a Piezo knockout, Piezo knockdown does not alter Albumin-FITC uptake. The authors should test the efficiency of their knockdown (using Piezo:GFP) and use a second, independent RNAi line to knockdown Piezo (if available).

We performed additional experiments to confirm the knockdown efficiency of our Piezo-RNAi strain. Of note, the same strain has been used by us and others before, to deplete Piezo in nephrocytes (1, 2). We could not generate stable flies expressing Piezo:GFP, *sns*-Gal4 and the RNAi strain due to time restrictions. A second RNAi strain would be available from VDRC, however, repeating all experiments is very time consuming, hence we decided to use a different approach to prove knockdown efficiency.

We recently established a fluorescence *in situ* hybridization (FISH) protocol in our lab and acquired Piezo-specific probes using the HCR™ Gold RNA-FISH tool from Molecular Instruments. By doing so, we have been able to visualize Piezo-specific mRNA in nephrocytes. Counting FISH signals (Piezo-specific mRNA) revealed a significant decrease in Piezo depleted nephrocytes when compared to controls. This data is now included in Supp. Figure 1A.

Based on these findings, we concluded that the knockdown is efficient and sufficient.

6. Fig. 3. The authors claim that, due to the temperature-sensitivity of the Gal4 system, Piezo over-expression is stronger at 29C compared with 25C. Since the authors express a FLAG-tagged Piezo, the authors should directly test this assumption by anti-FLAG immunostaining.

As suggested, we assessed expression levels by staining for the FLAG tag. We indeed observed a significant increase of FLAG intensity from 25 to 28°C. We performed the experiments on the same day, using the same antibody and kept laser intensity equal during imaging. We also show a control, which has no FLAG staining. The data is now part of Supp. Figure 2. We also added a citation for the temperature-sensitivity of the Gal4 system (3).

Supp. Fig. 2

7. Fig. 4E,F. The authors claim that Piezo depletion or overexpression results in a significant number of acidic vesicles. However, vesicles are difficult to detect in the images; moreover, the authors quantified mean intensities and not number of vesicles positive for lysotracker. The authors should attempt to acquire higher resolution images and quantify number of lysotracker positive vesicles.

As suggested, we have re-done and re-imaged the lysotracker staining. By doing so we acquired higher resolution images and used FIJI to count vesicles positive for lysotracker and bigger than 500nm in diameter. The analysis as well as new representative images are now included in Figure 4E,F. We observed a significant increase in the number of vesicles positive for lysotracker. We kept our conclusion, as counting vesicles resulted in the same effect as measuring intensity and is in line with the increased FITC-Albumin uptake in both genotypes. We also added a description of the analysis in the material and methods section.

8. Line 274. The authors speculate that this increase of acidic vesicles might be a consequence of extensive FITC-Albumin uptake. However, to the understanding of this reviewer, no FITC-Albumin was used in these experiments.

We agree, the wording of our conclusion was misleading. We changed this accordingly to:

This increase might be a result of extensive uptake of particles from the haemolymph – similar to the observed increase of FITC-Albumin uptake, which is subsequently degraded within nephrocytes.

9. Fig. 4G,H. The authors should mention what is shown in the plots (mean {plus minus}s.e.m?). Also, the authors should state the number of replicates analyzed.

We added the information in the figure legend. Depicted are means plus SEM. Each timepoint represents 3 independent n's with 15-20 1st instar larvae each.

10. Fig. 5A. The authors compare surface and cortical F-actin levels from control and Piezo overexpressing animals and report a significant difference. To test whether this difference is specific to F-actin, the authors should stain for a surface marker and a cortical marker (e.g. E-cadherin).

We agree this would be a nice additional piece of information. However, in our hands, phalloidin, which is a difficult stain in nephrocytes anyway, works best if used alone and no additional antibody-based staining is used. The phalloidin intensity of surrounding tissue is very high, making it difficult to image actin fibers in nephrocytes. Attempts to use antibodies in parallel resulted in less phalloidin intensity and thus difficulties in analysis. We therefore did not repeat our phalloidin stainings with additional markers.

11. Line 493. The conclusion that "...our findings show [...] that Piezo plays an important role in larval garland nephrocyte..." does not seem to be justified given the "mild functional phenotype" (see Abstract) of Piezo depletion.

We toned down our conclusion and took out this part of the sentence and rephrased this part of the discussion.

Reviewer #2 (Comments to the Authors (Required)):

This manuscript by Hazelton-Cavill et al. explores the role of the mechanotransducer Piezo in Drosophila nephrocytes as a model for mammalian podocytes. Using genetic manipulation (knockout, knockdown, and overexpression) and functional assays, the authors demonstrate that elevated Piezo levels cause nephrocyte injury associated with actin stress fiber accumulation, mitochondrial activation, oxidative stress, and increased Cubilin expression. They also show that short-term inhibition of Piezo with GsMTx4 partially rescues the phenotype. The study aims to contribute to understanding renal mechanotransduction and the pathological role of Piezo channels.

The topic of mechanosensitive ion channels in renal injury is timely and relevant. The use of Drosophila nephrocytes as a model system is well justified and aligns with previous work from the group. However, some points need further evaluation.

We thank the reviewer for the valuable assessment of our manuscript and the provided comments. We replied to all of them in detail below:

Major Comment

1. Could the authors show the level of Piezo downregulation achieved with the RNAi construct in garland nephrocytes? Demonstrating the level of knockdown, for example through quantitative PCR, immunostaining, or reporter signal reduction, would considerably strengthen the interpretation of the RNAi experiments.

We performed additional experiments to confirm the knockdown efficiency of our Piezo-RNAi strain. Of note, the same strain has been used by us and others before, to deplete Piezo in nephrocytes (1, 2). We could not perform Q-PCR, as we are not able to isolate nephrocytes and other tissue might also express Piezo, thus we might not be able to measure a decrease of Piezo in the knockdown nephrocytes. Also, we could not generate stable flies expressing Piezo:GFP, *sns-Gal4* and the RNAi strain, to test reporter signal reduction, due to time restrictions. We do not have a Piezo-specific Drosophila antibody and thus decided against the immunostaining approach, but we recently established a fluorescence *in situ* hybridization (FISH) protocol in our lab and generated Piezo-specific probes using the HCR™ Gold RNA-FISH tool from Molecular Instruments. Using this approach, we have been able to visualize Piezo-specific mRNA in nephrocytes. Quantification of Piezo-specific mRNA (FISH signals) revealed a significant decrease in Piezo depleted nephrocytes when compared to controls. This data is now included in Supp. Figure 1A.

Based on these findings, we concluded that the knockdown is efficient and sufficient.

2. *It would be helpful if the authors could indicate what proportion of garland nephrocytes normally express Piezo under physiological conditions. If expression is limited to a subset of cells, overexpression might induce Piezo in cells that are normally negative, potentially altering the phenotype.*

We addressed this comment together with the point below.

3. *The authors might wish to discuss whether Piezo-positive nephrocytes could represent a distinct subpopulation with specific structural or functional properties, different from Piezo-negative cells. Addressing this point would provide a more nuanced understanding of the cell type-specific effects of Piezo signalling.*

We have put together data from 8 Piezo-GFP MIMIC flies showing Piezo-GFP and Duf (used as nephrocyte marker) expression. We used Duf and GFP antibodies to visualize both proteins. Our data shows, that all cells positive for Duf also show GFP expression at the cell cortex. We thus concluded, that the majority of cells express Piezo under physiological conditions. Of note, we had to use an anti-GFP antibody to enhance the Piezo-GFP signal. This might be a result of low expression levels of Piezo under physiological conditions. Also, GFP levels differ between animals, which could be caused by antibody binding efficiency, as Duf levels also vary between flies. We decided to not include the data in the manuscript, but wanted to show our results here.

Also, by performing FISH with Piezo-specific mRNA probes in control flies (+; *sns-Gal4/w*;+) and wildtype (OreR) under physiological conditions (kept at 25°C, no additional treatment), we did not observe major differences between cells or between animals. All cells we analyzed did show Piezo-specific mRNA, the variation is considered biological variance between cells and animals. No control or wildtype animal had cells without any Piezo-specific mRNA. Again, we did not include the data in the manuscript, but wanted to show our data here in the response letter.

However, we did add a sentence, stating that the majority of cells express Piezo, in the manuscript.

4. Finally, I would encourage the authors to expand the discussion on the context- and cell type-specific roles of Piezo channels.

We expanded the discussion as recommended. Changes are marked in red.

Minor Comment

5. Colour for Duf in Figure1, panel E and F is unfortunate

We have re-done imaging with a better resolution and to address comments regarding localization at the cell cortex. These images now have different colors and we also do not show Pyd in the magnification to make it easier to distinguish between channels in the merge.

6. Fig 5 C, double bar in Piezo WT+ Rho DN

We apologize for this mistake and removed the second bar.

Reviewer #3 (Comments to the Authors (Required)):

Hazelton-Cavill and colleagues investigated the mechanosensitive ion channel Piezo in Drosophila garland cell nephrocytes. Exposing these cells to pressure reduced FITC-albumin endocytosis. Loss of Piezo did not affect the nephrocyte diaphragm but increased tracer uptake. High-level Piezo overexpression caused a mild reduction in nephrocyte diaphragms and also enhanced endocytosis of FITC-albumin and Cubilin, as well as LysoTracker labeling, actin, and mitochondrial markers. The phenotypes observed are mostly mild or non-significant, and the investigation is somewhat lacking in structure and well-supported conclusions. The strong emphasis on excessive overexpression further weakens the work. Two previous studies, including one from the authors' own laboratory, have already examined Piezo function in pericardial nephrocytes. Overall, this study offers only rather modest novel biological insight.

We thank the reviewer for the valuable comments regarding our manuscript and replied to all comments below:

Comments:

1. I am not convinced of the physiological relevance of the pressure-stimulus experiments. Applying 0.5 bar (50,000 Pa) seems orders of magnitude above biologically plausible values. For comparison, shear stress in capillaries is around 0.1-5 Pa. Would a freely floating garland nephrocyte ever experience forces of this magnitude, or are the authors simply inducing mechanical cell damage?

We agree that this *ex vivo* experiment might not resemble physiological conditions as experienced by floating nephrocytes. However, we think this a reasonable experiment and setup as described below. In our setup, the pressure generated to release the buffer is given as 0.5 bar by the pressure injector, however there is a loss of pressure which is experienced by nephrocytes. The needle is approximately 1cm away from the glass dish and the surface of the buffer in which the dissected tissue is floating. Unfortunately, we are not able to measure the pressure in the glass dish and experienced by nephrocytes. We agree that it is likely the mechanical stimulus induced here is higher than forces *in vivo*. However, we have used this approach before to study Ca⁺⁺ influx in the absence and presence of Yoda1 as well as effects of Yoda1 on nephrocyte biology (1). In these experiments we also applied a mechanical stimulus to control cells (DMSO) and did not observe a morphological or functional phenotype in larval garland nephrocytes. These data suggest that even if the pressure is higher than *in vivo*, we do not induce severe damage through mechanical stress.

2. The *Piezo::EGFP Mi{MIC}* line was already characterized by Zhao et al., showing localization in channels. The present manuscript claims localization "close to the nephrocyte diaphragm" without providing adequately magnified cross-sections to support a conclusion divergent from Zhao et al.

We apologize if our conclusion was phrased in a misleading way, we do not propose a different localization from that described previously by Zhao et al.. We are really careful with the way in which we comment on localization based on immunofluorescence images, as we do not have any additional data, such as immunoprecipitation to claim Piezo expression at the nephrocyte diaphragm. To our understanding, Figure 1 in the Zhao et al manuscript shows UAS-Piezo-GFP in combination with the novel Sns-Ruby fly strain. The overexpression of Piezo results in expression in the lacunae, which we also observe in our overexpression flies as seen in Supplementary Figure 2A,C and E. We also observed a finger-print like pattern in the overexpression lines. In addition, Zhao et al see and describe in their manuscript: 'Using another *Piezo-GFP* reporter line [*Piezo::EGFP Mi{MIC}*], in which the endogenous *Piezo* is tagged by *GFP*, we confirmed the same cortical fingerprint pattern (Supplemental Figure 1, A and B).' We used the same *Piezo-GFP* MIMIC line from BDSC and also observed a fingerprint like pattern. We

have re-done imaging of Piezo-GFP MIMIC flies with a better resolution and to visualize the cortical area (Figure 1).

We apologize, if our conclusion was not phrased clearly enough, but from our point of view, we have similar findings in regard to Piezo localisation in wildtype (MIMIC) and overexpression flies as to what has been reported for adult pericardial cells by Zhao et al. We thus rephrased our conclusion in the manuscript. Changes are marked in red.

3. How is the increase in FITC-albumin uptake upon both loss and overexpression of Piezo explained, when only overexpression results in higher Cubilin levels?

We agree with the reviewer, this is an interesting finding, which we cannot explain at the moment. Our intention behind using the Cubilin antibody was to identify a possible mechanism explaining the increased FITC-Albumin uptake. However, the whole-body knockout of Piezo does not show alterations in Cubilin expression. In the discussion we added some additional thoughts in regard to this finding:

Additionally, we cannot exclude the involvement of other components of the nephrocyte uptake machinery such as Amnionless, which was not assessed within this project. Other downstream pathways including endocytosis and degradation processes might also be altered upon loss of Piezo, impacting on FITC-Albumin processing once it has been taken up.

4. Lysotracker broadly labels different acidic compartments. It remains unclear what this indicates or how it relates to the phenotype. How is increased Lysotracker „a consequence of extensive FITC-albumin uptake“?

We agree, lysotracker dyes accumulate in acidic organelles or vesicles and thus label lysosomes, late endosomes and other acidic vesicles. Thus, by using lysotracker we cannot distinguish between those vesicles. To further assess, which vesicles are altered upon depletion and overexpression of Piezo, we also analyzed Rab7, a late endosome marker. This did not reveal significant differences in nephrocytes expressing Piezo WT or in Piezo^{KO} cells. This finding suggests that changes observed in lysotracker positive vesicles are primarily not mediated by late endosomes. We kept the term ‘acidic vesicles’ in the manuscript, as we still cannot distinguish between lysosomes and other acidic vesicles, and as lysotracker has been previously used in *Drosophila* to identify organelle acidification (2, 6–8). The Rab7 data is now part of Supp. Figure 4D.

We agree, the wording of our conclusion was misleading. We changed this accordingly to:

This increase might be a result of extensive uptake of particles from the haemolymph – similar to the observed increase of FITC-Albumin uptake, which is subsequently degraded within nephrocytes.

5. Piezo overexpression increases endocytosis but diminishes toxin resistance. This contradictory finding remains unexplained. Does nephrocyte function improve or worsen?

We agree, one would expect a beneficial effect on pupal development upon overexpression of Piezo due to enhanced FITC-Albumin uptake. Here we show evidence of increased Cubilin, which has been previously reported to be important for Albumin uptake into cells (mammalian and fly). It is possible that nephrocytes filter/ take up factors from the haemolymph with a higher rate upon over-expression of Piezo and this might impact on development. These might also include non-toxic, beneficial factors, which are needed for development. We thus deliberately did not use the terms improved or worsen function, as the alterations observed could be a worsening of function in both cases. We included our thoughts mentioned here in the manuscript and kept our previous thoughts on this in the results section:

This effect on haemolymph clearance is surprising, given that overexpression of Piezo results in elevated uptake of FITC-Albumin, potentially mediated via greater accessibility of Cubilin and increased Cubilin expression. Although one might expect a similar elevation in AgNO₃ uptake upon Piezo overexpression, since AgNO₃ can also be taken up by Cubilin (9), additional effects of Piezo overexpression in nephrocytes might influence pupal development, as this is delayed already under basal conditions without feeding AgNO₃. The AgNO₃ toxin assay also differs from the FITC-Albumin uptake assay in that it can be considered a long-term readout, assessing uptake function over days, whilst the FITC-Albumin uptake assay is a

short-term read-out, focusing on uptake within minutes only. Also, it is possible that nephrocytes do filter/ take up factors including AgNO_3 from the haemolymph with a higher rate upon over-expression of Piezo. However, this might also include the uptake of non-toxic or beneficial factors, which are needed in the haemolymph for normal development. Thus, differences in effects of FITC-Albumin uptake and clearance function/developmental delay might be observed here.

6. Higher mitochondrial abundance could follow a long list causes beyond cytosolic calcium, which was not even shown in this study.

Based on literature we decided to assess mitochondria and ROS formation in relation to the overexpression of a calcium channel. We agree, we do not provide data showing that calcium is causative for higher abundance of mitochondria, but we also do not claim this link. Our suggestion of increased calcium levels in nephrocytes overexpressing Piezo is based on previous findings using Yoda1. Zhao et al. showed an increase of intracellular calcium levels in nephrocytes after Yoda1 treatment (specific Piezo activator). Enhancing the amount of Piezo channels could thus also result in increased intracellular calcium and might impact on mitochondria and ROS. As we do not provide data confirming this link, we only conclude an effect on mitochondria by overexpression of Piezo, but do not mention any mechanistic link.

7. Why wasn't the combination of Rho DN and Piezo consistently controlled for GAL4 dilution within the same experiment?

There was no specific reason for this. We did the GFP dilution experiment at a later timepoint to rule out dilution effects.

8. GsMTx4 hardly seems a plausible candidate for a "novel therapeutic treatment in Piezo associated kidney disease." The latter is not even an established clinical entity.

GsMTx4 has been used in other animal studies, in which it proved to be beneficial as it improved kidney function in a lupus nephritis mouse model, and in the tubulointerstitial fibrosis model in mice with unilateral ureter obstruction (UUO) or with folic acid treatment. These studies are mentioned in the discussion. Based on these published findings and our own data, we concluded GsMTx4 could be a useful treatment option in the future, after additional testing. Also in other research areas such as Duchenne muscular dystrophy GsMTx4 has been assessed in mouse models to block Piezo and validate the toxin as a future therapeutic (10).

9. *The authors should more thoroughly discuss their findings in the context of previous studies on Piezo in nephrocytes (ER stress? Autophagy?).*

We added info about the manuscript by Zhao et al in the discussion.

10. *Some sections of the manuscript seem slightly wordy and at times repetitive.*

We rephrased some parts of the manuscript.

References

1. **Schulz K, Hazelton-Cavill P, Alornyo KK, Edenhofer I, Lindenmeyer M, Lohr C, Huber TB, Denholm B, Koehler S.** Piezo activity levels need to be tightly regulated to maintain normal morphology and function in pericardial nephrocytes. *Sci Rep* 14: 28254, 2024. doi: 10.1038/s41598-024-79352-9.
2. **Zhao Y, Duan J, Han ID, Van De Leemput J, Ray PE, Han Z.** Piezo, Nephrocyte Function, and Slit Diaphragm Maintenance in *Drosophila*. .
3. **Duffy JB.** GAL4 system in *drosophila* : A fly geneticist's swiss army knife. *Genesis* 34: 1–15, 2002. doi: 10.1002/gene.10150.
4. **Hermle T, Braun DA, Helmstädter M, Huber TB, Hildebrandt F.** Modeling Monogenic Human Nephrotic Syndrome in the *Drosophila* Garland Cell Nephrocyte. *J Am Soc Nephrol* 28: 1521–1533, 2017. doi: 10.1681/ASN.2016050517.
5. **Dow JAT, Simons M, Romero MF.** *Drosophila melanogaster*: a simple genetic model of kidney structure, function and disease. *Nat Rev Nephrol* 18: 417–434, 2022. doi: 10.1038/s41581-022-00561-4.
6. **Zhu J, Lee J-G, Fu Y, Van De Leemput J, Ray PE, Han Z.** *APOL1-G2* accelerates nephrocyte cell death by inhibiting the autophagy pathway. *Disease Models & Mechanisms* 16: dmm050223, 2023. doi: 10.1242/dmm.050223.
7. **Fu Y, Zhu J, Richman A, Zhang Y, Xie X, Das JR, Li J, Ray PE, Han Z.** *APOL1-G1* in Nephrocytes Induces Hypertrophy and Accelerates Cell Death. *JASN* 28: 1106–1116, 2017. doi: 10.1681/ASN.2016050550.
8. **Lee J-G, Fu Y, Zhu J, Wen P, Van De Leemput J, Ray PE, Han Z.** A SNARE protective pool antagonizes *APOL1* renal toxicity in *Drosophila* nephrocytes. *Cell Biosci* 13: 199, 2023. doi: 10.1186/s13578-023-01147-8.
9. **Zhang F, Zhao Y, Chao Y, Muir K, Han Z.** Cubilin and Amnionless Mediate Protein Re-absorption in *Drosophila* Nephrocytes. *JASN* 24: 209–216, 2013. doi: 10.1681/ASN.2012080795.
10. **Wang W, Huang M, Huang X, Ma K, Luo M, Yang N.** GsMTx4-blocked *PIEZO1* channel promotes myogenic differentiation and alleviates myofiber damage in Duchenne muscular dystrophy. *Skeletal Muscle* 15: 13, 2025. doi: 10.1186/s13395-025-00383-5.

January 23, 2026

RE: Life Science Alliance Manuscript #LSA-2025-03515-TR

Dr. Sybille Koehler
Universität Hamburg
Martinistr. 52
Hamburg 20257
Germany

Dear Dr. Koehler,

Thank you for submitting your revised manuscript entitled "Elevated Piezo levels cause structural and functional alterations in *Drosophila* garland nephrocytes".

Your manuscript was evaluated by all the original reviewers. As you will note, your revisions addressed most of their concerns. However Reviewer 3 notes that the manuscript description misses a statement on the limitation in one of the experimental approaches, i.e. in using unphysiological force on the cell.

We agree that you must respond to this concern in the manuscript text while describing the referenced results by explicitly stating (as you have done on page 17 of your response letter) that (1) the ex-vivo experiment does not resemble physiological conditions experienced by floating nephrocytes and (2) the mechanical stimulus induced by your experimental set-up is likely higher than the forces in-vivo due to loss of pressure.

Overall, we would be happy to publish your paper in Life Science Alliance pending resolution of the above point and final revisions necessary to meet our formatting guidelines.

MANUSCRIPT ORGANIZATION AND FORMATTING:

To avoid unnecessary delays in the acceptance and publication of your paper, please read the following information carefully. Full guidelines are available on our Instructions for Authors page, <https://www.life-science-alliance.org/authors>

- In your abstract, we suggest that you make the connection between fly nephrocytes and mammalian podocytes in a brief manner.
- For Figures 1 , 4, Suppl Figure 4, please provide a scale bar and size information for magnified views.
- Please provide a citation for fly food recipe or provide a detailed composition.
- Please complete details in different imaging sections to include objective type, N.A. and step size for imaging. Please also include an imaging section following the description of LysoTracker/MitoSox/Mitotracker/Actin fibre staining.
- Thank you for providing citations for the FITC-Albumin Uptake assay. Kindly provide some more details on the assay as applied in this study.
- Please provide the sequence for the Piezo Probe using in FISH experiments
- Kindly confirm the genotypes stated in legend for Figure 2, in particular for kd.
- Please include callouts in manuscript text for: Figure 6E; Supplementary Figures 2A-B, 3A-F, 4A-B, 5A-B.
- Please remove List of Supplementary data from main manuscript file.
- Please leave only one clean version of manuscript file in .docx file format.
- Please upload Supporting figures as individual files without captions included. Supporting figure captions should be placed in manuscript file after the main figure captions.
- Please consult our manuscript preparation guidelines <https://www.life-science-alliance.org/manuscript-prep> and make sure your manuscript sections are in the correct order.
- Please use the [10 author names, et al.] format in your references (i.e. limit the author names to the first 10).
- Please add the X and Bluesky handles of your host institute/organization as well as your own or/and one of the authors in our system.
- Please be sure that the authorship listing and order is correct.

It is Life Science Alliance policy that if requested, original data images must be made available to the editors. Failure to provide

original images upon request will result in unavoidable delays in publication. Please ensure that you have access to all original data images prior to final submission.

LSA encourages authors to provide a 30-60 second video where the study is briefly explained. We will use these videos on social media to promote the published paper and the presenting author (for examples, see <https://docs.google.com/document/d/1-UWCfbE4pGcDdcgzcmiuJl2XMBJnxKYeqRvLLrLSo8s/edit?usp=sharing>). Corresponding or first-authors are welcome to submit the video. Please submit only one video per manuscript. The video can be emailed to contact@life-science-alliance.org

FINAL FILES:

The following items are required for acceptance.

The license to publish form must be signed before your manuscript can be sent to production. A link to the license to publish form will be available to the corresponding author only. Please take a moment to check your funder requirements.

Thank you for your attention to these final processing requirements. Please revise and format the manuscript and upload materials as soon as you are able.

Thank you for this interesting contribution to the literature. We look forward to publishing your paper in Life Science Alliance.

Sincerely,

Sarita Hebbar, PhD
Scientific Editor
Life Science Alliance
<http://www.lsajournal.org>

Reviewer #1 (Comments to the Authors (Required)):

The authors have satisfactorily addressed my comments.

Reviewer #2 (Comments to the Authors (Required)):

The authors strongly improved the manuscript and answered all my previous questions. It is now acceptable for publication

Reviewer #3 (Comments to the Authors (Required)):

The revised manuscript includes additional controls, which improve the study and some concerns are addressed. Unfortunately,

the fundamental weakness of the manuscript remains that a series of loosely connected effects are described on the basis of excessive overexpression. This contrasts with the strong wording of the interpretations.

The fundamental limitation of applying unphysiological forces on the cell by the pressure stimulus (Fig. 1) should be mentioned in this otherwise rather wordy manuscript.

January 27, 2026

RE: Life Science Alliance Manuscript #LSA-2025-03515-TRR

Dr. Sybille Koehler
Universität Hamburg
Martinistr. 52
Hamburg 20257
Germany

Dear Dr. Koehler,

Thank you for submitting your Research Article entitled "Elevated Piezo levels cause structural and functional alterations in *Drosophila* garland nephrocytes". It is a pleasure to let you know that your manuscript is now accepted for publication in Life Science Alliance. Congratulations on this interesting work.

DISTRIBUTION OF MATERIALS:

Again, congratulations on a very nice paper. I hope you found the review process to be constructive and are pleased with how the manuscript was handled editorially. We look forward to future exciting submissions from your lab.

Sincerely,

Sarita Hebbar, PhD
Scientific Editor
Life Science Alliance
<http://www.lsjournal.org>